



# Implications of present ground temperatures and relict stone stripes in the Ethiopian Highlands for the palaeoclimate of the tropics

Alexander R. Groos[1], Janik Niederhauser[1], Luise Wraase[2], Falk Hänsel[2], Thomas Nauss[2], Naki Akçar[3], and Heinz Veit[1]

[1]Institute of Geography, University of Bern, 3012 Bern, Switzerland
[2]Department of Geography, Philipps University of Marburg, 35032 Marburg, Germany
[3]Institute of Geological Sciences, University of Bern, 3012 Bern, Switzerland

**Correspondence:** Alexander R. Groos (alexander.groos@giub.unibe.ch)

**Abstract.** Large sorted patterned grounds are the most prominent features of periglacial and permafrost environments of the mid and high latitudes, but have not yet been verified for the tropics. Here, we report on relict large sorted polygons (up to 8 m in diameter) and large sorted stone stripes (up to 1000 m long, 15 m wide, and 2 m deep) on the ~4000 m high Sanetti Plateau in the Bale Mountains, southern Ethiopian Highlands. For a systematic investigation of past and present frost-related

processes and landforms in the Bale Mountains, we conducted geomorphological mapping both in the field and on satellite images. The sorted stone stripes were studied in more detail by applying aerial photogrammetry, ground-penetrating radar measurements, and [36]Cl surface exposure dating. In addition, we installed 29 ground temperature data loggers between 3493 and 4377 m to analyse present frost occurrence and seasonal temperature variations from 2017 to 2020. Finally, we ran a simple experiment and combined recent ground temperature measurements with meteorological data in a statistical model to

assess the air temperature depression needed for the past formation of deep seasonal frost and cyclic freezing and thawing on the plateau. Our results show that relict and modern periglacial landforms are common in the Bale Mountains. Nocturnal superficial ground frost on the plateau occurs at 35-90 days per year, but the mean annual ground temperature (~11 °C) is far off from seasonal or permanent frost conditions. The modelling experiment suggests a minimum air temperature depression on the plateau of 7.6 ± 1.3 °C for the emergence of several decimetre deep seasonal frost. The stone stripes probably formed

under periglacial conditions in proximity of a palaeo ice cap on the plateau during the coldest period(s) of the last glacial cycle. We hypothesise that the slightly inclined and unglaciated areas of the plateau, the coexistence of regolith and large blocks, the occurrence of deep seasonal frost, as well as relatively dry conditions beyond the ice cap provided ideal conditions for frost heave and sorting and the formation of large sorted patterned grounds. The presence of these landforms and the associated air temperature depression provide further evidence for an amplified cooling of high tropical mountains during the last glacial

period that is yet not well captured in global climate models.

## 1 Introduction

The Earth experienced a pronounced global-mean cooling of 5-7 °C during the global Last Glacial Maximum (LGM; 22 ± 4 ka after Shakun and Carlson, 2010) compared to the pre-industrial climate, but the magnitude of cooling and expansion of



ice sheets and glaciers varied considerably across the globe. While global climate models point towards a maximum cooling over the northern hemisphere ice sheets, they suggest only a moderate cooling for the tropics (Schneider von Deimling et al., 2006). A synthesis of sea surface temperatures testifies to a moderate mean zonal LGM cooling of the tropical oceans of ~2 °C (MARGO Project Members, 2009). However, terrestrial temperature reconstructions based on calculated snow line depressions

on tropical mountains indicate a much stronger and more heterogeneous cooling of 2-14 °C (Mark et al., 2005). Other terrestrial palaeoclimate data from the low-latitudes confirm the pronounced cooling over land and highlight an amplified cooling with increasing elevation (Farrera et al., 1999; Loomis et al., 2017). Disentangling the causes for the mismatch between the reconstructed cooling of the tropical oceans and land areas, especially at high elevation, is crucial for understanding and simulating global climate changes during the last glacial period because the energy excess in the the tropics and transport towards

higher latitudes drives the large-scale atmospheric and oceanic circulation (Kageyama et al., 2005). Model uncertainties and limitations as well as erroneous marine and terrestrial temperature reconstructions have been discussed as potential causes for the discrepancy. Climate model experiments and latest temperature reconstructions along an elevational gradient in Eastern Africa provide an alternative explanation: they support the interpretation that the amplified cooling at high elevations in the tropics during the global LGM was the result of a drier atmosphere and steeper lapse rate (Kageyama et al., 2005; Tripati et al.,

2014; Loomis et al., 2017). However, climate proxy data from the high tropical mountains are still sparse although they are essential for quantifying global LGM temperature changes in the middle troposphere (Farrera et al., 1999).

A promising region for high-elevation palaeoclimatic and geoecological reconstructions in the tropics are the Bale Mountains in the southern Ethiopian Highlands as they comprise Africa's largest alpine environment and provide manifold evidence for past glacial and periglacial processes (Grab, 2002; Osmaston et al., 2005; Hendrickx et al., 2014; Groos et al., in revision).

During the Late Pleistocene, an ice cap with several outlet glaciers covered the central Sanetti Plateau and northern valleys of the Bale Mountains. The local maximum glacier expansion in the region was reached between 50-30 ka, well before the global LGM, and coincided with a temperature depression of 4-6 °C (Groos et al., in revision). In view of the gradual global cooling until 22 ± 4 ka, an even stronger temperature depression (>6 °C) in the Ethiopian Highlands seems likely after 50-30 ka. A conspicuous geomorphological feature beyond the glacial remains of the former ice cap on the Sanetti Plateau are large

sorted stone stripes (several meters wide and hundred meters long) between 3850 and 4150 m. They are associated with past sporadic permafrost and might indicate a severe cooling in the Bale Mountains during the Pleistocene (Miehe and Miehe, 1994; Grab, 2002). Sorted patterned grounds of similar size are typical for periglacial and permafrost environments of the mid and high latitudes (Goldthwait, 1976; André et al., 2008). Diurnal freeze-thaw cycles in tropical mountains are sufficient for the development of small-scale patterned grounds (Francou et al., 2001), but the large dimension of the stone stripes on the

Sanetti Plateau is unique for the low latitudes as their formation presumably requires deep ground frost and seasonal freezing and thawing (e.g. Kessler and Werner, 2003). A systematic investigation of the relict periglacial landforms and present frost patterns in the Bale Mountains is lacking. When, how, and under which environmental conditions the relict patterned grounds formed and what their occurrence implies for the palaeoclimate of the tropics is therefore still unexplored.

The aim of this study is the systematic investigation of past and present frost-related processes and landforms in the Bale Moun-

tains to elaborate the potential of geomorphological features like the large sorted stone stripes for paleoclimatic reconstructions





at high-elevations in the tropics. For gaining insights into the spatial and elevational distribution of relict and active periglacial landforms, we conducted geomorphological mapping both in the field and on high-resolution satellite images. The sorted stone stripes on the Sanetti Plateau were studied in more detail by applying aerial photogrammetry, ground-penetrating radar (GPR) measurements, and $^{36}$Cl surface exposure dating. The $^{36}$Cl ages were originally published by Groos et al. (in revision) in a

palaeoglaciological context, but we present them here again as they are also of relevance for the interpretation of the relict stone stripes. Since knowledge on present frost occurrence and ground temperature variations in the Bale Mountains is indispensable for discussing how and under which climatic and environmental conditions the relict structures may have formed, we installed a ground temperature network covering the Sanetti Plateau and northeastern declivity. In a final step, we combined the ground temperature measurements with meteorological data from nearby weather stations and applied a simple statistical

model to assess the minimum air temperature depression needed for the formation of deep frost and patterned grounds in the tropical Ethiopian Highlands.

## 2  Study area

The Bale Mountains (6.6–7.1 °N, 39.5–40.0 °E) are located southeast of the Main Ethiopian Rift and belong to the Bale-Arsi massif which constitutes the western part of the southern Ethiopian Highlands (Fig. 1). Precambrian rocks and overlying

Mesozoic marine sediments form the base of the massif and are covered by Cenozoic trachytic and basaltic lava flows (Miehe and Miehe, 1994; Osmaston et al., 2005; Hendrickx et al., 2014). Due to the lack of geological maps, lithological information, geochemical studies, and radiometric dating, especially in the southern Ethiopian Highlands, the exact timing of volcanic eruptions is unknown and the successive formation of the Bale-Arsi massif still poorly understood (Mohr, 1983; Osmaston et al., 2005). Characteristic for the Bale Mountains is the central Sanetti Plateau with a mean elevation of ~4000 m. It is

bounded to the west by hardly weathered lava flows, to the north and east by broad U-shaped valleys, and to the south by the Harenna Escarpment. Several volcanic plugs and cinder cones like the highest peak Tullu Dimtu (4377 m) rise above the plateau (Osmaston et al., 2005). With an area of almost 2000 km² above 3000 m, the Bale Mountains comprise Africa's most extensive alpine environment (Groos et al., in revision) and are an important fresh water source for the surrounding lowlands. The main tributaries of the only two perennial rivers in the Somali lowlands, Shebelle and Jubba, originate from the Bale

Mountains.

The seasonal movement of the Intertropical Convergence Zone (ITCZ) and zonal shift of the Congo Air Boundary, which defines the confluence of air masses from the Indian Ocean and Atlantic, determines the climate and rainfall patterns of the Ethiopian Highlands (Levin et al., 2009; Tierney et al., 2011; Costa et al., 2014). Due to the complex topography, the mean annual precipitation varies considerably across the region and is strongly linked to elevation (Gebrechorkos et al., 2019). Three

seasons characterise the current climate: The dry season (traditionally called "Bega") lasts from November to February and is followed by two rainy seasons ("Belg" and "Kiremt"). While "Belg" (March to June) is more pronounced in the southern Ethiopian Highlands, "Kiremt" (July to October) plays a major role in the northern highlands including the upper catchment area of the Blue Nile (Conway, 2000; Seleshi and Zanke, 2004). During the dry season, when the ITCZ in Eastern Africa is

Earth **Surface**
**Dynamics**
Discussions

**Figure 1.** Overview map of the experimental setup and observation network in the Bale Mountains, southern Ethiopia. The lowest weather stations Magnete (1599 m) and Delo Mena (1315 m) are located 25 and 40 km south of Rira. GT: high-quality temperature data loggers. TM: low-cost temperature data loggers. Rock samples: blocks sampled for surface exposure dating. Control points: natural objects used for the georeferencing of the high-resolution orthophoto and digital surface model. GPR: ground-penetrating radar. Data basis: SRTM 1 Arc-Second Global (United States Geological Survey) and high-resolution WorldView-1 satellite image (DigitalGlobe Foundation).



located south of the equator and high pressure cells have established over Western Asia and the Sahara, northeasterly trade winds from the Arabian Peninsula and Sea prevail in the Bale Mountains and cause only little precipitation. Along with the northward movement of the ITCZ from March to June, the main wind direction changes from northeast to southeast and brings moist air from the southern Indian Ocean to the Bale Mountains (Lemma et al., 2020). Although the Gulf of Guinea and

Congo Basin are important moisture sources for the northern Ethiopian Highlands (Levin et al., 2009; Viste and Sorteberg, 2013; Costa et al., 2014), they seem of minor relevance for the Bale-Arsi massif (Lemma et al., 2020). The Sanetti Plateau and highest peaks of the massif experience occasional snowfall during the rainy seasons, but the thin snowpack usually melts within hours or days (Miehe and Miehe, 1994).

Like most of the other tropical mountains in Eastern Africa, the Bale Mountains are currently unglaciated. The present mean

0 °C isotherm (a rough proxy for the modern snow line in the tropics) is located at least 300 m above the highest peak Tullu Dimtu. However, latest glacial geomorphological and chronological studies provide clear evidence that the snow line was much lower during the Late Pleistocene and favoured the formation of an extensive plateau glaciation with outlet glaciers extending down into the northern valleys. Between 50-30 ka during the local Last Glacial Maximum in the southern Ethiopian Highlands, ice covered about 265 km² of the Bale and additional 83 km² of the adjacent Arsi Mountains. Two later glacial stages were dated

to ~18 and ~15 ka. At ~18 ka, the ice extent was slightly smaller than during the local LGM, but ice still covered the central part of the plateau and northern valleys. Deglaciation in the region set in after ~18-15 ka. The highlands remained probably ice-free over the entire Holocene (Ossendorf et al., 2019; Groos et al., in revision). Besides glacial landforms like moraines and roche moutonnées, also relict periglacial features have been reported from the Bale Mountains (Grab, 2002; Hendrickx et al., 2014). Among those, large sorted stone circles (several meters in diameter) and stone stripes (several meters wide and hundred meter

long) on the Sanetti Plateau are the most prominent ones. The formation of such large features is associated with freeze-thaw processes and indicates decimetre to meter deep seasonal frost or sporadic permafrost with a thick active layer (see Sections 3.7, 4, and 5). Field observations and short-term ground temperature measurements between December 1989 and March 1990 verify that nocturnal frost near the soil-atmosphere interface still occurs. The most apparent results of the modern freeze-thaw cycles are the formation of needle ice along saturated stream banks and the presence of sorted stone nets in flat and poorly

drained areas on the Sanetti Plateau (Miehe and Miehe, 1994; Grab, 2002).

Afroalpine herbs, grasses, *Helichrysum* dwarf shrubs, extrazonal patches of *Erica*, and giant lobelias cover the ice-free and barren Sanetti Plateau today (Miehe and Miehe, 1994). The plateau is home to many endemic plant species like the giant lobelia (*Lobelia rhynchopetalum*) (Chala et al., 2016) and mammal species like the Ethiopian wolf (*Canis simensis*) (e.g. Gottelli et al., 1994), giant mole-rat (*Tachyoryctes macrocephalus*) (e.g. Vlasatá et al., 2017), and mountain nyala (*Tragelaphus*

*buxtoni*) that are restricted to the Ethiopian Highlands (Miehe and Miehe, 1994). Since these endemic species mainly populate the upper valleys and Sanetti Plateau today, palaeoclimatic and -environmental changes like a severe cooling, expansion of the ice cover and periglacial area as well as depression of altitudinal vegetation belts must have directly affected their habitat and are therefore also of relevance in a geoecological context.



## 3 Data and methods

### 3.1 Mapping of periglacial landforms

Comprehensive and thorough geomorphological mapping of glacial and periglacial landforms provides crucial data for establishing glacial chronologies and reconstructing the palaeoclimate and palaeoenvironment of polar and alpine regions (Chandler

et al., 2018). We evaluated maps, photographs, and field notes from previous studies dealing with periglacial processes and landforms in the Bale Mountains (e.g. Messerli and Winiger, 1992; Miehe and Miehe, 1994; Grab, 2002; Umer et al., 2004; Osmaston et al., 2005) to compile evidence of relict and modern frost occurrence. Since periglacial landforms have yet not been described systematically, we performed extensive geomorphological mapping on the Sanetti Plateau, along the upper Harenna Escarpment and in the western, northern, and eastern valleys during multiple field excursions between 2016 and 2020 (Fig.

2). In addition, we evaluated high-resolution WorldView-1 satellite images (pixel size = 0.5 m) provided by the DigitalGlobe Foundation to identify features in remote and difficult-to-access areas of the mountain range. All periglacial features discovered in the field or on satellite images were geotagged and compiled in a catalogue (see Appendix A).

For a detailed analysis of the geomorphology, geometry, and size of the sorted stone stripes on the Sanetti Plateau, we conducted a manual aerial survey (~50 m above ground level) with a small quadcopter (DJI Mavic Pro) on the 30th January 2020

at 2 pm local time. A total of 75 aerial images were acquired during the survey and processed with the photogrammetric software OpenDroneMap (following the approach described in Groos et al., 2019) to obtain a high-resolution orthophoto (5 cm) and digital surface model (DSM, 10 cm) of the stone stripes. Five natural objects (rocks and dwarf shrubs) visible on the orthorectified WorldView-1 image and at least on three aerial images were used as ground control points (Fig. 1) for a rough georeferencing of the orthophoto and DSM (Groos et al., 2019). The necessary elevation information were extracted from the

SRTM 1 Arc-Second Global dataset.

### 3.2 Ground-penetrating radar measurements

Investigating the internal structure of the sorted stone stripes by excavating a transect was not possible due to regulations. Instead, we performed a ground-penetrating radar (GPR) survey between two stripes on the southern Sanetti Plateau on the 10th February 2020 (Fig. 1). We made use of the Pulse EKKO PRO GPR with a 1000 MHz antenna (7.5 cm sensor width)

manufactured by Sensors & Software Inc., which was originally purchased by another subproject of the Ethio-European DFG Research Unit 2358 "The Mountain Exile Hypothesis" for geoecological investigations (for system settings see Appendix B). The GPR was mounted on a compatible pushcart (Fig. 2). As survey setting, an exploration depth of 1 m and pulse length of 16 nanoseconds (ns) was applied for the first line and modified to 1.5 m depth and 24 ns pulse length for the following lines. The starting point of the GPR measurement was located 10 m above the position of data loggers GT07-09 since the uppermost

part of the volcanic plug was not accessible with the pushcart. Due to uneven terrain and several natural obstacles like smaller stones and *Helichrysum* dwarf shrubs, the GPR profile between the two stone stripes was split into five separate lines varying between 3.8 and 38.5 m in length. The chaotic structure of the stones stripes prevented a GPR survey inside the troughs and coarse material. For analysis and visualisation of the GPR data, we used the software EKKO Project (version 5.0).





**Figure 2.** Field work in the Bale Mountains: (a) reconnaissance and mapping of periglacial landforms, (b) sampling of stone stripes for surface exposure dating, (c-e) installation of ground temperature loggers, and (f) ground-penetrating radar survey.

### 3.3  $^{36}$Cl surface exposure dating of periglacial landforms

Previous studies have demonstrated that the stabilisation age of relict periglacial features like rock glaciers and block fields can be successfully dated with cosmogenic nuclides (e.g. Barrows et al., 2004; Ivy-Ochs et al., 2009; Steinemann et al., in press). In analogy, we sampled two sorted stone stripes on the Sanetti Plateau, one about 5 km south and another one about 10 km west of Tullu Dimtu (Fig. 1), to determine the stabilisation age of these features. The results were originally published by Groos et al. (in revision) in a palaeoglaciological context. We present them here briefly again as they are also of relevance for the discussion of the origin of the sorted stone stripes. From both stone stripes, we selected three columnar rocks for $^{36}$Cl surface exposure dating (Table 1). To avoid distorting effects on exposure dating due to strong shielding in the trough-shaped stripes or toppling of rocks after the stabilisation phase, we only chose rocks that were sticking out a bit and were wedged between other rocks. The upper few centimetres of each target rock were sampled with hammer, chisel, and angle grinder for



**Table 1.** Description of periglacial features on the Sanetti Plateau sampled for [36]Cl surface exposure dating.

| Rock sample | Lithology | Latitude (°N) | Longitude (°E) | Elevation (m a.s.l.) | Boulder length (m) | Boulder width (m) | Boulder height (m) | Sample thickness (cm) | Shielding factor |
|---|---|---|---|---|---|---|---|---|---|
| BS01 | Basalt | 6.78634 | 39.79297 | 3874 | 2.1 | 0.6 | 1.0 | 2.5 | 0.9961 |
| BS02 | Basalt | 6.78660 | 39.79280 | 3869 | 1.5 | 0.5 | 1.4 | 4.5 | 0.9961 |
| BS03 | Basalt | 6.78682 | 39.79263 | 3865 | 0.6 | 0.4 | 1.0 | 3.0 | 0.9997 |
| BS04 | Basalt | 6.85491 | 39.72078 | 4050 | 0.8 | 0.6 | 1.1 | 5.0 | 0.9990 |
| BS05 | Trachyandesite | 6.85513 | 39.72074 | 4049 | 0.5 | 0.5 | 1.0 | 4.5 | 0.9990 |
| BS06 | Trachyandesite | 6.85550 | 39.72049 | 4045 | 1.5 | 0.5 | 0.6 | 3.5 | 0.9994 |

Data from Groos et al. (in revision).

the subsequent laboratory analysis (Fig. 2). An inclinometer was used in the field for measuring the topographic shielding. For extraction of the isotope [36]Cl, the six samples were crushed, sieved and chemically treated in the Surface Exposure Dating Laboratory of the University of Bern. Total Cl- and [36]Cl-concentrations (see Appendix C) were measured from one target at the 6 MV AMS-facility of the ETH Zurich using the isotope dilution technique (Ivy-Ochs et al., 2004) and a gas-filled magnet

to separate [36]S (Vockenhuber et al., 2019). For a detailed description of the sample preparation, Cl and [36]Cl measurements, and surface exposure age calculation see Groos et al. (in revision).

### 3.4 Ground temperature measurements

For measuring hourly ground temperatures of the Bale Mountains, we installed high-quality UTL-3 Scientific Dataloggers (hereafter GT data loggers) in 2, 10, and 50 cm depth at five different locations with little vegetation between 3877 and 4377 m

(Fig. 1 and Table 2). The GT data loggers are developed by GEOTEST Ltd. in collaboration with the Swiss Institute for Snow and Avalanche Research. They consist of a waterproof housing, a YSI 44005 thermistor for measuring temperature, a memory for up to 65.000 readings (>7 years by hourly interval), a replaceable 3.6 V lithium battery for power supply, and a USB interface for data transfer. According to the manufacturer, the measurement accuracy is <0.1 °C at 0 °C and the thermometric drift per 100 months is <0.01 °C at 0 °C. At each of the five measurement sites, the upper 50 cm of the ground were removed

to install the GT data loggers (Fig. 2). We used data loggers with an external cable and thermistor for the measurements in 10 and 50 cm depth. A standard logger without external cable was placed just below the surface in 2 cm depth. After the installation, each hole was filled in the same order as during the excavation to ensure as little disturbance of the profile as possible. Additional low-cost tempmate.-B2 temperature data loggers (hereafter TM data loggers) in the size of a button cell (Fig. 2) were distributed across the Bale Mountains between 3493 and 4377 m to increase the spatial coverage of near-surface

(2 cm) hourly ground temperature measurements (Fig. 1 and Table 2). The single-use TM data loggers consist of a splashproof housing (we wrapped the loggers in thin tape for better protection), an unspecified thermistor, a memory for up to 8192 readings (341 days by hourly interval), and an irreplaceable 3.0 V battery. A logger-pan-to-USB cable is needed for connecting





**Table 2.** Overview of the installed ground temperature data loggers (excluding six lost items).

| Data logger | Latitude (°N) | Longitude (°E) | Elevation (m a.s.l.) | Depth (cm) | Slope (°) | Aspect (°) | Start of measurement | Readout dates |
|---|---|---|---|---|---|---|---|---|
| GT16 | 6.92725 | 39.77275 | 4153 | 2 ± 1 | 22 | 140 | 31.12.17 | 14.06.18, 23.01.20 |
| GT02 | 6.92725 | 39.77275 | 4153 | 10 ± 2 | 22 | 140 | 06.01.17 | 17.12.17, 31.12.17, 14.06.18, 23.01.20 |
| GT03 | 6.92725 | 39.77275 | 4153 | 50 ± 5 | 22 | 140 | 06.01.17 | 17.12.17, 31.12.17, 14.06.18, 23.01.20 |
| GT17 | 6.93000 | 39.77188 | 4181 | 2 ± 1 | 19 | 35 | 31.12.17 | 14.06.18, 23.01.20 |
| GT05 | 6.93000 | 39.77188 | 4181 | 10 ± 2 | 19 | 35 | 06.01.17 | 17.12.17, 31.12.17, 14.06.18, 23.01.20 |
| GT06 | 6.93000 | 39.77188 | 4181 | 50 ± 5 | 19 | 35 | 06.01.17 | 17.12.17, 31.12.17, 14.06.18, 23.01.20 |
| GT07 | 6.78665 | 39.79342 | 3877 | 2 ± 1 | 8 | 320 | 21.01.17 | 10.12.17, 06.01.18, 25.01.20 |
| GT08 | 6.78665 | 39.79342 | 3877 | 10 ± 2 | 8 | 320 | 21.01.17 | 10.12.17, 06.01.18, 25.01.20 |
| GT09 | 6.78665 | 39.79342 | 3877 | 50 ± 5 | 8 | 320 | 21.01.17 | 10.12.17, 06.01.18, 25.01.20 |
| GT10 | 6.79474 | 39.81469 | 3932 | 2 ± 1 | 10 | 130 | 21.01.17 | 11.12.17, 06.01.18, 26.01.20 |
| GT11 | 6.79474 | 39.81469 | 3932 | 10 ± 2 | 10 | 130 | 21.01.17 | 11.12.17, 06.01.18, 26.01.20 |
| GT12 | 6.79474 | 39.81469 | 3932 | 50 ± 5 | 10 | 130 | 21.01.17 | 11.12.17, 06.01.18, 26.01.20 |
| GT13 | 6.82617 | 39.81897 | 4377 | 2 ± 1 | 0 | - | 21.01.17 | 19.12.17, 20.01.20, 26.01.20 |
| GT14 | 6.82617 | 39.81897 | 4377 | 10 ± 2 | 0 | - | 21.01.17 | 19.12.17, 20.01.20 |
| GT15 | 6.82617 | 39.81897 | 4377 | 50 ± 5 | 0 | - | 21.01.17 | 19.12.17, 26.01.20 |
| TM04 | 6.84411 | 39.87876 | 4129 | 2 ± 1 | 0 | - | 18.01.17 | 09.12.17, 05.01.18, 10.06.18 |
| TM05 | 6.77522 | 39.80307 | 3858 | 2 ± 1 | 0 | - | 18.01.17 | 09.12.17, 06.01.18, 29.12.18, 25.01.20 |
| TM06 | 6.77535 | 39.80311 | 3857 | 2 ± 1 | 0 | - | 18.01.17 | 09.12.17, 06.01.18, 29.12.18 |
| TM07 | 6.77521 | 39.80318 | 3856 | 2 ± 1 | 0 | - | 18.01.17 | 09.12.17, 06.01.18, 29.12.18, 25.01.20 |
| TM08 | 6.82617 | 39.81897 | 4377 | 2 ± 1 | 0 | - | 21.01.17 | 19.12.17 |
| TM09 | 6.86644 | 39.74365 | 4084 | 2 ± 1 | 0 | - | 23.01.17 | 12.12.17, 15.06.18, 24.01.20 |
| TM10 | 6.85509 | 39.71345 | 4022 | 2 ± 1 | 0 | - | 23.01.17 | 13.12.17, 15.06.18, 24.01.20 |
| TM11 | 7.01307 | 39.72272 | 3493 | 2 ± 1 | 0 | - | 29.12.17 | 14.06.18 |
| TM12 | 6.95493 | 39.73463 | 3769 | 2 ± 1 | 0 | - | 30.12.17 | 14.06.18, 22.01.20 |
| TM13 | 6.91937 | 39.76898 | 3930 | 2 ± 1 | 0 | - | 31.12.17 | 14.06.18, 22.01.20 |
| TM14 | 6.82605 | 39.80496 | 4124 | 10 ± 2 | 0 | - | 06.01.18 | 30.12.18, 26.01.20 |
| TM15 | 6.81928 | 39.81152 | 4185 | 10 ± 2 | 0 | - | 06.01.18 | 30.12.18, 16.02.20 |
| TM16 | 6.81327 | 39.81968 | 4103 | 10 ± 2 | 0 | - | 06.01.18 | 30.12.18, 26.01.20 |
| TM17 | 6.79197 | 39.81005 | 3880 | 10 ± 2 | 0 | - | 06.01.18 | 31.12.18, 26.01.20 |

the TM loggers to a computer and retrieving the data. The measurement accuracy is ± 0.5 °C at -10 to 65 °C according to the manufacturer. Due to the much lower accuracy of the TM data loggers compared to the GT data loggers, we performed a comparative measurement indoor over several hours with logger GT04 as reference. Since the root-mean-square deviation

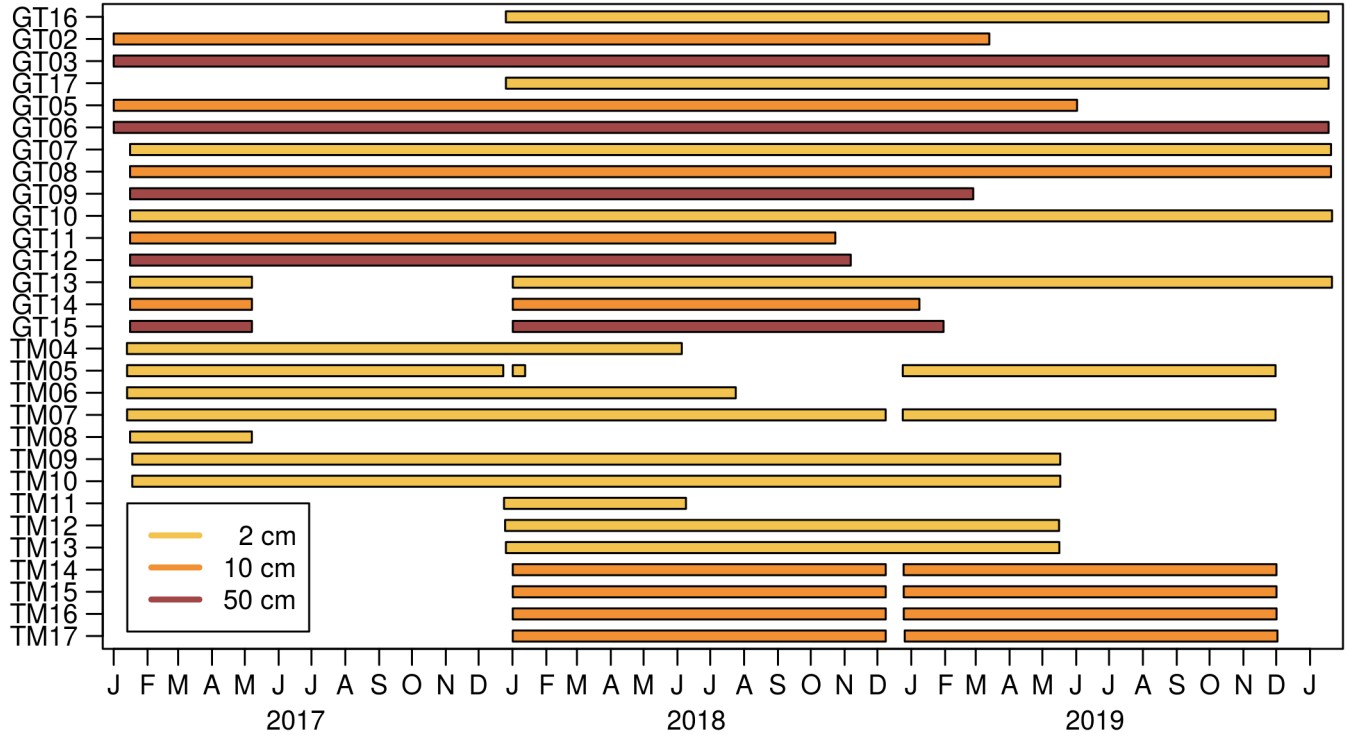

**Figure 3.** Measurement period(s) of each ground temperature data logger between January 2017 and January 2020. For the statistical gap-filling method applied to obtain complete time series see Section 3.6.

of each TM data logger from the reference measurement was smaller than the stated accuracy of ± 0.5 °C, a calibration was unnecessary. For a direct cross-comparison in the field, data logger TM08 was installed next to GT13 in 2 cm depth on top of Tullu Dimtu. Two TM data logger (16 and 17) were buried below *Erica* in 10 cm depth for comparison with sites (TM14 and TM15) with only little vegetation.

5    Several issues occurred during the measurement period from January 2017 to January 2020 and caused considerable data gaps (Fig. 3). The data logger positions were orginially marked with four small plastic poles in 2 m distance. They were apparently too conspicuous and led to the loss of several items (GT01, GT04, GT18, TM01-03, TM08, and TM11). Therefore, we removed all poles and switched to natural markers (dwarf shrubs, stones, etc.). Data loggers GT01 and GT04 were replaced by GT16 and GT17. On Tullu Dimtu, data loggers GT13-15 were taken away in May 2017, but could be recovered and reinstalled in January

10    2018. Furthermore, we also noticed a relatively short battery life of just two years for some of the GT and TM data loggers leading to a substantial data loss. Two years are much shorter than the battery life stated by both manufacturers for the respective sampling interval (GT ~ 3-5 years, TM ~ 5 years). After consultation, the manufacturer of the GT data loggers adjusted the internal handling of the lithium batteries to ensure the stated battery life. The two data loggers TM05 and TM06 temporarily recorded unrealistic low values (down to -40 °C). Individual outliers and longer periods with implausible measurements were



**Table 3.** Overview of the installed automatic weather stations in the Bale Mountains.

| Weather station | Location | Latitude (°N) | Longitude (°E) | Elevation (m a.s.l.) | First measurement | Last measurement | Data completeness (%)* |
|---|---|---|---|---|---|---|---|
| BALE001 | Tullu Dimtu | 6.82693 | 39.81871 | 4377 | 04.02.17 | 31.01.20 | 73 |
| BALE002 | Tuluka | 6.78945 | 39.77511 | 3848 | 02.02.17 | 30.01.20 | 100 |
| BALE003 | Angesso Station | 6.89642 | 39.90854 | 3949 | 31.01.17 | 30.01.20 | 68 |
| BALE004 | Magnete | 6.51622 | 39.74515 | 1599 | 06.02.17 | 01.02.20 | 100 |
| BALE005 | Meskel Darkina | 7.05860 | 39.62336 | 3724 | 09.02.17 | 05.10.19 | 97 |
| BALE006 | Rira Substation | 6.75912 | 39.72161 | 2803 | 06.02.17 | 19.02.20 | 86 |
| BALE007 | Dinsho Head Quarter | 7.09378 | 39.78966 | 3208 | 28.01.17 | 20.02.20 | 100 |
| BALE008 | Sodota** | 6.99249 | 39.70171 | 3529 | 29.01.17 | 21.04.18 | 100 |
| BALE009 | EWCP Station | 6.84945 | 39.88197 | 4124 | 01.02.17 | 30.01.20 | 100 |
| BALE010 | Delo Mena | 6.41199 | 39.83328 | 1315 | 11.02.17 | 01.02.20 | 100 |

*Ratio of actual to maximum possible measurements during the respective measurement period. **The weather station was abondoned after breakdown in April 2018.

removed from the time series. After the first reading of data logger GT07, we realised that it was accidentally placed in 6 cm depth and not in 2 cm as intended. The relocation towards the surface in December 2017 led to an abrupt increase in the temperature amplitude. Therefore, we calculated hourly ground temperature gradients between 6 and 10 cm depth from GT07 and GT08 data by applying a simple linear regression to extrapolate the GT07 measurements from 6 to 2 cm in the period

5   21st January to 10th December 2017. A compilation of all ground temperature time series from the Bale Mountains is provided in Table S1. Data gaps were filled using a simple linear regression and data from other GT or TM data loggers to generate a complete data set for the period 1st of February 2017 to 20th January 2020 (see Section 3.6). All data modifications made for each logger are listed in Table S2 (see data availability statement for supplementary Tables S1-4).

### 3.5   Meteorological measurements

10   Within the framework of the DFG Research Unit 2358 ten automatic weather stations (AWS) were installed in the Bale Mountains national park between 1315 and 4377 m beginning of 2017 (Table 3). The AWS are manufactured by Campbell Scientific and consist of a three metre galvanised tubing tripod, a grounding kit, a weather-resistant enclosure, a measurement and control system (CR800), a solar module (SDT200), a 168 Wh battery, a charging regulator, a temperature and relative humidity probe (CS215) with radiation shield, a pyranometer (LI-200R), a two-dimensional ultrasonic anemometer from Gill

15   Instruments, and a rain gauge from Texas Electronics (TR-525USW 8"). For protection, the AWS are wire-fenced by a 3 x 3 m compound. Air temperature, relative humidity, and global radiation are measured in 2 m height, wind speed and wind direction in 2.6 m height, and precipitation in 1 m height. The measurement interval is 15 minutes. All measured variables are finally aggregated to hourly averages.





## 3.6    Statistical data interpolation and analysis

To obtain a complete and consistent data set of hourly ground temperatures for the Bale Mountains from 1[st] February 2017 until 20[th] January 2020 (Table S3), we applied a statistical gap-filling approach. Most of the ground temperature measurements from different locations or depths overlap for a certain period in time (see Fig. 3) and allow to establish a statistical relationship.

We applied a simple linear regression model to interpolate missing data points in the time series of a logger using data from a nearby logger for which measurements were available. If multiple logger with a similar equidistance came into question for the interpolation, we chose the one that yielded the best fit (evaluated by the coefficient of determination $R^2$) and lowest root-mean-square error (RMSE). The overlapping measurement period between the predicting logger and dependent logger was split into a calibration and validation part. For the interpolation of incomplete time series in 10 or 50 cm depth, we drew

on available data from 2 cm depth of the same location. We used a moving average of the data from 2 cm depth to account for the time-lag response to atmospheric changes in greater depths. The number of preceding hours considered for the calculation of the moving average that yielded the best prediction (high $R^2$, low RMSE) of the ground temperatures in 10 or 50 cm depth was chosen. Details on the data gap-filling of each incomplete time series are provided in Table S4. The validation of the simple linear models applied for interpolation revealed an average $R^2$ of 0.86 ± 0.07 and RMSE of 1.9 ± 1.4 °C. The time

series of the data loggers TM05-06, TM08, and TM14-17 were not interpolated because the data served only for comparative experiments (low-cost vs. high-quality loggers, vegetated vs. unvegetated locations, etc.) and were dispensable for the temporal and elevational analysis.

Most of the AWS installed in the Bale Mountains measured continuously, but some of the time series are interrupted due to issues with the power supply (Table 3). The hourly meteorological data from the different AWS are stored in an online database

and gaps in the time series of all variables except wind speed and direction are interpolated statistically following a workflow developed by Wöllauer et al. (in revision). Single missing values are interpolated linearly using the average of the adjacent data points. Longer gaps in a time series are filled using available data from several nearby AWS. A multiple linear regression model fitted with data from the overlapping measurement periods is applied to predict the missing values from data of those AWS that reveal a strong correlation and low RMSE. Predictor variables (AWS) with a high $R^2$ and low error are given a higher

weight in the interpolation.

We evaluated the interpolated hourly meteorological and ground temperature data statistically to quantify frost occurrence and spatio-temporal ground temperature variations in the Bale Mountains. Twelve data loggers from 2 cm depth (excluding TM05-06, TM08, and TM14-17) and five loggers from 10 and 50 cm depth were considered for calculating mean annual ground temperatures, daily ground temperature cycles, thermal gradients, number of frost days, frost penetration depth, elevational

gradients, etc. To study seasonal ground temperature variations related to changes in insolation, cloudiness and humidity, we conducted the calculations separately for the entire study period, the dry season (Bega: November – February), and the two rainy seasons (Belg: March – June, Kiremt: July – October). Furthermore, comparative measurements were performed to investigate the differences in ground temperature between north- and south-facing slopes (GT16 and GT02-03 vs. GT17 and





GT05-06), vegetated and unvegetated areas (TM16-17 vs. TM14-15), and the performance of low-cost vs. high-quality data loggers (TM08 vs. GT13).

## 3.7 Ground temperature modelling and palaeoclimate reconstruction

The potential of periglacial landforms for paleoclimatic and environmental reconstructions has already been pointed out in

pioneering studies from more than half a century ago (e.g. Galloway, 1965). Periglacial landforms are often more abundant than glacial deposits, especially in dry regions, and can be a more reliable climate proxy than palaeo glacier extents as their formation is less sensitive to changes in precipitation. Here, we explore a novel and experimental approach to infer palaeoclimatic information from relict periglacial landforms and established ground temperature modelling (e.g. MacLean and Ayres, 1985) using on-site meteorological data and present ground temperature measurements.

We make the following main assumptions for our model experiment: First, the large sorted stone stripes on the Sanetti Plateau are of periglacial origin and their formation required deep seasonal frost or sporadic permafrost with a thick active layer (see Sections 4 and 5 for arguments supporting this interpretation). Second, deep seasonal frost (or permafrost) forms when the long-term mean annual ground temperature is < -1 °C. Third, the impact of the geothermal heat flux on ground temperatures near the surface is negligible in the Bale Mountain. The principal idea of the introduced method is to establish a statistical rela-

tionship between the measured ground temperatures and a set of meteorological variables for simulating under which climatic conditions (e.g. decrease in air temperature and insolation) the mean ground temperature would approximate frost conditions. For the development of separate multiple linear regression models, we considered three locations on the Sanetti Plateau where ground temperatures and meteorological variables were measured simultaneously (Tullu Dimtu, EWCP Station, Tuluka). We chose only air temperature and global radiation as explanatory variables. The wind speed time series contains data gaps, pre-

cipitation is limited to individual rain events, and relative humidity does not show a direct linear relationship with ground temperature (see Section 4.3). The multiple linear regression model at each site was calibrated for the period 1st February 2017 – 31st January 2019 and validated for the period 1st February 2019 – 20th January 2020. Based on the established statistical relationship, present-day hourly ground temperatures in 2 cm ($T_{2cm}$) can be modelled using measured air temperature and incoming shortwave radiation:

$$T_{2cm,i} = \beta_0 + (\beta_1 \times T_{air,i}) + (\beta_2 \times Q_{S,i}), \qquad (1)$$

where $T_{air,i}$ ($i = 1,\dots,n$) is the hourly measured air temperature in °C, $Q_{S,i}$ is the hourly measured incoming shortwave radiation in W m$^{-2}$, $\beta_0$ is the intercept, $\beta_1$ is the coefficient for $T_{air}$, and $\beta_2$ is the coefficient for $Q_S$. The coefficients and goodness of fit for each of the three linear models are provided in Table 4. For simulating past ground temperatures, two additional parameters, $\Delta T_{air}$ and $\Delta Q_S$, were introduced:

$$T_{2cm,i} = \beta_0 + (\beta_1 \times (T_{air,i} - \Delta T_{air})) + (\beta_2 \times (Q_{S,i} - \Delta Q_S)), \qquad (2)$$

where $\Delta T_{air}$ is the difference between the mean present-day and past air temperature in °C and $\Delta Q_S$ is the difference between the mean present-day and past incoming shortwave radiation in W m$^{-2}$. For simplicity, we set $\Delta Q_S$ to 30 W m$^{-2}$ (the rough





**Table 4.** Coefficients and goodness of fit of the three established multiple linear regression models (MLRM) with ground temperature as dependent and air temperature and global radiation as explanatory variables. Distance means the distance between AWS and data logger, $\beta_0$ is the intercept, $\beta_1$ the air temperature coefficiet, and $\beta_2$ the incoming shortwave radiation coefficient.

| Linear regression model | Elevation (m) | Distance (m) | $\beta_0$ | $\beta_1$ | $\beta_2$ | R² cal | RMSE cal (°C) | R² val | RMSE val (°C) |
|---|---|---|---|---|---|---|---|---|---|
| MLRM Tullu Dimtu | 4377 | 90 | 3.7 | 1.7 | 0.004 | 0.73 | 3.0 | 0.72 | 3.0 |
| MLRM EWCP Station | 4124 | 690 | 1.2 | 1.6 | 0.010 | 0.79 | 3.6 | 0.76 | 3.6 |
| MLRM Tuluka | 3848 | 2050 | -0.5 | 1.9 | 0.004 | 0.63 | 4.9 | 0.78 | 4.0 |

lowering of incoming shortwave radiation during the LGM at 15 °N, see Groos et al., in revision). To infer the air temperature depression at the formation time of the periglacial landforms using *Eq. 2*, we increased $\Delta T_{air}$ (starting with: $\Delta T_{air} = 0$ °C) with every iteration until $T_{2cm}$ became $< -1$°C. We tested all three developed multiple linear regression models (Tullu Dimtu, EWCP Station, and Tuluka) to quantify the uncertainty of the approach originating from differences in the model coefficients 5 $\beta$ (Table 4). Since the lowest-situated stone stripes on the Sanetti Plateau are located at an elevation of 3870-3890 m, we used meteorological data ($T_{air}$ and $Q_S$) from the Tuluka AWS at 3848 m to run the three models. Alternatively, the meteorological data from the higher-situated AWS (Tullu Dimtu and ECWP Station) can be adjusted to the elevation of the stone stripes using a lapse rate of 0.7 °C per 100 m (see Section 4.2). Running each model with the locally adjusted meteorological data led to the same calculated temperature depression as using the Tuluka AWS data. We rescaled the simulated ground temperatures in 2 10 cm depth to the maximum seasonal ground temperature variations in 10 and 50 cm depth observed today (see Section 4.2) to model temperature variations in these depths:

$$T_{50\text{cm,i}} = (\overline{T}_{2\text{cm}} - a) + \frac{(T_{2\text{cm,i}} - min(T_{2\text{cm}})) \times (b-a)}{(max(T_{2\text{cm}}) - min(T_{2\text{cm}}))}, \tag{3}$$

where ($T_{50cm,i}$) are the simulated ground temperatures in 50 cm depth in °C ($i = 1,\dots,n$), $\overline{T}_{2cm}$ is the mean air temperature in 2 cm depth in °C, $a$ (= - 1.25 °C) is the predefined seasonal minimum, and $b$ (= 1.25 °C) the predefined maximum of $T_{50cm,i}$. For 15 10 cm depth ($T_{10cm}$), $a$ equals to -3 °C and $b$ to 3 °C. The main drawback of the presented approach is the non-consideration of ground moisture and thermal conductivity due to the lack of respective measurements. To further improve the method in the future, profile sensors measuring moisture, electrical conductivity, and temperature in 5 cm intervals between 0 and 50 cm depth have been installed at three AWS on the Sanetti Plateau in January 2020. The data are not yet available.

## 4 Results

20 ### 4.1 Distribution and characteristics of periglacial landforms

The Bale Mountains comprise a wide range of periglacial landforms and other characteristic geomorphological features related to present and relict frost occurrence (Fig. 4 and Appendix A). Current frost-induced phenomena like frozen waterfalls, needle ice, patterned grounds, and solifluction lobes are limited to the upper part of the valleys (>3900 m), Sanetti Plateau, and highest

Earth **Surface**
**Dynamics**
Discussions
EGU

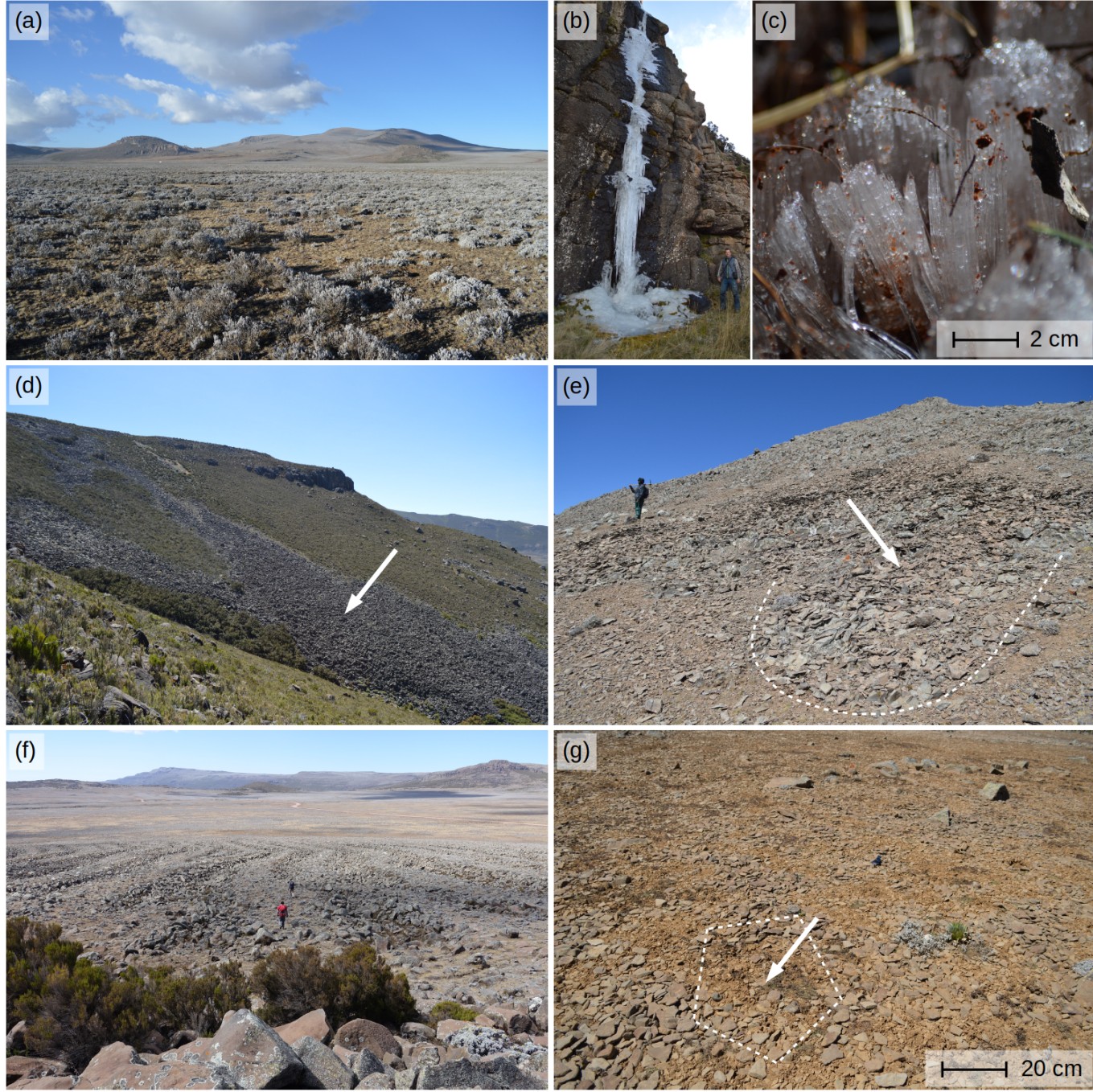

**Figure 4.** Periglacial environment of the Bale Mountains: (a) view from the southern Sanetti Plateau towards Tullu Dimtu, (b) frozen waterfall and (c) needle ice in the Wasama Valley, (d) relict blockfields along the southern Harenna Escarpment, (e) active solifluction lobes at Mt. Wasama, (f) relict sorted stone stripes, and (g) active sorted polygons on the Sanetti Plateau.




Earth **Surface**
**Dynamics**
Discussions
**Figure 5.** (a) Overview map of periglacial landforms and other characteristic geomorphological features in the Bale Mountains. (b-c) Sorted stone stripes in the western (b) and southern part (c) of the Sanetti Plateau as seen on WorldView-1 satellite images (DigitalGlobe Foundation). (d) High-resolution orthophoto and DSM cross-section profile of the stone stripes derived from the aerial images.

peaks, even though ground temperatures below freezing can sporadically extend to much lower elevations (down to 2700-3000 m). We observed needle ice (3-5 cm long) mainly along water-saturated stream banks at places with cold air ponding. Needle ice is a typical small-scale example for diurnal freeze-thaw cycles in the Bale Mountains as it forms at clear nights through-

5 out the dry season. Interestingly, we also found evidence for seasonal frost phenomena. Up to 10 m high frozen water falls evolve at shaded north-exposed cliffs in the Wasama Valley during the dry season and last until the onset of the following wet season. Active small-scale polygonal stone nets occur in flat and poorly drained areas on the Sanetti Plateau and unvegetated solifluction lobes above 4100 m at the southern slopes of Mount Wasama. Compared to the modern periglacial processes and landforms, relict geomorphological features are larger and much more pronounced in the Bale Mountains.

Most of the relict periglacial features can be found along the Harenna Escarpment, on the Sanetti and Genale Plateau, and

10 at the slopes of the highest peaks (Fig. 5a). Characteristic for the highest peaks of the northern declivity are bare and gentle





slopes and the accumulation of scree below heavily eroded basaltic and trachytic cliffs. This type of deposits is associated inter alia with frost weathering and differs from the chaotic spread of individual boulders below elongated cliffs at lower elevations. Another conspicuous landform associated with periglacial activity are large blockfields located between 3500 and 4000 m at the southern and western slopes of the Sanetti Plateau. The blockfields consist of hardly weathered angular boulders and are no

longer active as the presence of lichens and partly reoccupation by *Erica* prove. Circular patterns across the Sanetti and Genale Plateau as well as elevated areas of the northern declivity are not further considered here since they are, at least in some areas, of biogenic origin related to the activity of the endemic giant mole-rat (Miehe and Miehe, 1994). The most striking geomorphological features on the Sanetti Plateau are large sorted patterned grounds comprising stone stripes and less developed stone circles.

The large sorted stone stripes occur exclusively on the southern and western Sanetti Plateau and at one site on the lower Genale Plateau (Fig. 5). On the southern Sanetti Plateau and on the Genale Plateau, the stone stripes formed at gentle slopes (inclination: 2 – 9°) of three different volcanic plugs between 3700 and 3950 m. The stone stripes consist of hardly weathered angular or columnar basalt boulders (Fig. 2 and 6), are partly covered by lichens, and are up to 200 m long, 15 m wide, and 2 m deep (Fig. 5c). While the stone stripes are trough-shaped, the areas with finer material inbetween are more rampart-like (Fig. 5d).

The distance between the stone stripes equals in most cases to the width of the stripes. Typical for some of the stone stripes is that they split up into two narrower branches in the upper part and merge to a single wider branch in the lower part. As the GPR survey suggests, the regolith layer between the stone stripes contains no larger rocks (exceeding several decimetres) and is more than 1.5 m deep (6b). The surface of the underlying solid rock was not detected. All larger rocks (up to 0.5 m wide and 2 m long) are located mainly in the troughs or on top of the regolith layer. On the slightly inclined (2 - 9°) western

Sanetti Plateau between 3950 and 4150 m, the stone stripes are 300 – 1000 m long and mainly 5 – 10 m wide (Fig. 5b). Most of the stripes are connected to heavily eroded cliffs. In the upper part, some of the stripes split up into multiple branches and where the plateau flattens, a transition from sorted stone stripes to less developed stone circles is visible in the field, but hardly recognisable on satellite images.

The six dated rock samples from two different locations on the Sanetti Plateau originate from basaltic (BS01-04) and trachytic

(BS05-06) lava flows as it is indicated by the varying alkali and silica contents (Table C1). We obtained very high $^{36}$Cl concentrations, especially for the two trachytic samples (>120 × $10^6$ At g$^{-1}$) from the western part of the plateau (Table C2). In these two samples (BS05 and BS05), $^{36}$Cl has reached saturation. This means that the production and decay of $^{36}$Cl average out. Since the resulting exposure ages (>1000 ka) are at the limit of the method, they are not explicitly stated in the figures and tables. Based on the remaining samples, we calculated non-erosion-corrected $^{36}$Cl surface exposure ages of 84 ± 4, 281 ± 12,

and 281 ± 13 ka for the southern and of 620 ± 13 ka for the western stone stripes (Table C2). However, due to the high $^{36}$Cl concentrations, an erosion rate of >1 mm ka$^{-1}$ would lead to considerably older exposure ages for all samples except BS01. The "old" ages conflict with a relatively young formation age (e.g. global LGM or postglacial) as suggested by the morphology and hardly weathered surface of the investigated angular and columnar boulders. Long-term exposure of the sampled rocks to $^{36}$Cl-producing cosmic rays prior to the formation of the stone stripes could explain this mismatch. Despite the high $^{36}$Cl

concentrations, a temporary ice cover overlying the stone stripes for several thousand years during the last glacial cycle cannot

Earth **Surface**
**Dynamics**
Discussions

EGU

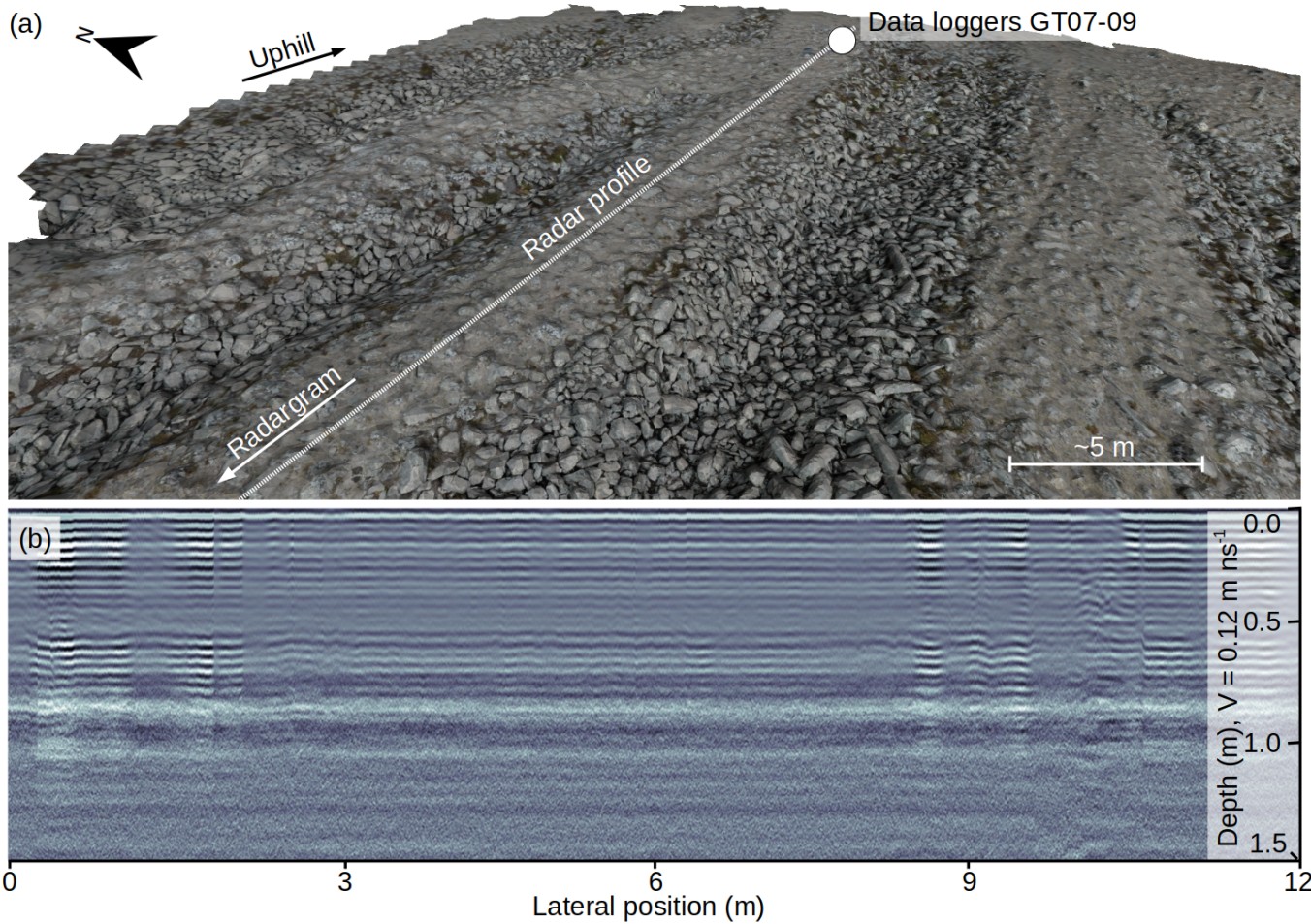

**Figure 6.** (a) 3D aerial view and (b) radargram of the sorted stones stripes on the southern Sanetti Plateau. For the location of the displayed radargram section (GPR05) see Fig. 1.

be entirely ruled out from the exposure dating alone. A meter-thick ice cover would reduce the production rate, but a period of several thousand years would not be sufficient to affect the $^{36}$Cl concentrations noticeably or zero the inheritance. However, a temporary ice cover overlying the stripes seems unlikely in light of the absent field evidence for such a scenario.

### 4.2 Present frost occurrence and ground temperature variations

5 The observed present-day ground temperatures in the Bale Mountains show characteristic daily and seasonal variations, but are in general way off from permafrost conditions (Fig. 7). At the location of the stone stripes on the southern Sanetti Plateau, the mean multiannual ground temperature between the surface and 50 cm depth is 11 °C. On top of the highest peak Tullu Dimtu, the mean annual ground temperature is 7.5 °C. The mean air temperature at the same location is 2 °C and therefore about 5.5



**Figure 7.** (a) Hourly ground temperatures and (b) seasonal ground temperature variations in 2, 10, and 50 cm depth on the southern Sanetti Plateau (3877 m) from January 2017 to January 2020. A local regression with a smoothing span of 0.32 was applied to derive seasonal ground temperature variations from hourly measurements. Daily mean ground temperature variations in 50 cm are provided additionally (thin dark red line).



K lower than the mean ground temperature. While the daily ground temperature range is largest near the surface and decreases with depth, the seasonal variations in all depths follow a similar cycle (Fig. 7). On the plateau, the ground cools down during the dry season (Bega) and warms up during the wet seasons (Belg and Kiremt). The difference between the seasonal minimum and maximum of daily mean temperatures over a year is about 10 K near the surface, 6 K in 10 cm, and 2.5 K in 50 cm depth (in

Fig. 7, daily mean ground temperatures are only presented for 50 cm depth). This shows that seasonal temperature variations can also be of relevance for tropical mountains with a pronounced diurnal climate. The time series is far too short for deriving any long-term trends, but the interannual ground temperature variability observed during the three-year period (2017 – 2020) was rather low (<0.5 °C).

Near the surface, the diurnal ground temperature amplitude is well pronounced and varies on average between 10-20 °C

during the wet and 20-30 °C during the dry season (Fig. 8a). Extreme temperatures of up to 45-50 °C during cloudless days and down to -10 °C during clear night have been observed on the Sanetti Plateau. Nocturnal ground frost on the plateau occurs at 35-90 days per year. However, the frost penetrates only the uppermost centimetres. The diurnal amplitude decreases considerably with increasing depth. At 10 cm depth, temperatures below freezing have not been measured at any of the logger locations during the entire study period. The annual ground temperature profile in the upper 50 cm is homogeneous. The

daily temperature difference between the surface and 50 cm depth is rarely larger than ± 2 °C (Fig. 8b). Annual ground temperatures increase from Tullu Dimtu down to the lowest logger location in the Web Valley by 0.71 °C per 100 m (Fig. 8c), but nocutural frost occurs in the valleys still at 5-25 days per year. The ground temperature gradient of 0.71 °C per 100 m is similar to the annual lapse rate obtained for the plateau (0.70 °C per 100 m) and northern declivity from measured air temperatures (Fig. 9). Interestingly, the lapse rate obtained for the Harenna Escparment is less steep (0.62 °C per 100 m) and

might represent wetter conditions and pronounced cloud formation at the southern declivity between ~1500-3800 m compared to the drier plateau and northern declivity. However, the number of AWS below the afro-alpine belt is not sufficient to determine unequivocally elevations where the mean annual lapse rate changes from a more or less dry adiabatic to a moist adiabatic and vice versa. Distinct elevational changes in the lapse rate could indicate the mean annual condensation level as well as the upper atmospheric storey where dry north-easterly trade winds dominate.

The comparative experiment on Mount Wasama shows clear differences between the thermal regime of the southern and northern slope (Fig. 10a). The southern slope is on average more than 2 °C warmer and reveals a more pronounced seasonality and larger diurnal amplitude which favours freezing and thawing and might explain the presence of solifluction lobes. While the mean daily temperature at the southern slope peaks towards the end of the dry season (January to February) when the sun is in its zenith, it reaches its maximum at the northern slope a few month later when the sun approaches its northernmost position.

The ground temperature differences between vegetated and unvegetated areas on the Sanetti Plateau are less obvious (Fig. 10b). Small *Erica* trees and bushes buffer the diurnal temperature amplitudes of the shaded ground, but have only little impact on the seasonality. Like at Mount Wasama, both south exposed locations at Tullu Dimtu (vegetated and unvegetated) have their temperature maxima at the end of the dry season, whereas the vegetated and unvegetated locations in the plain heat up during June/July. The cross-comparison between low-cost and high-quality data loggers on top of Tullu Dimtu revealed a promising

relationship ($R^2$ = 0.98) and proved that the tested low-cost loggers, which have not been developed explicitly for scientific

Earth **Surface**
**Dynamics**
Discussions

**Figure 8.** (a) Mean multiannual and multiseasonal diurnal ground temperature cycle in 2, 10, and 50 cm depth considering all data (as defined in Section 3.6) between February 2017 and January 2020 from the respective depths. The shaded areas display the mean diurnal ground temperature variability (as standard deviation) resulting from the different logger locations. First column: multiannual mean, second column: multiannual dry season mean (Bega: Nov/Dec/Jan/Feb), third column: multiannual wet season mean (Belg: Mar/Apr/May/Jun), fourth column: multiannual wet season mean (Kiremt: Jul/Aug/Sep/Oct). (b) Multiannual (black) and multiseasonal (Bega: green, Belg: blue, Kiremt: purple) daily ground temperature profiles in 2-50 cm depth between February 2017 and January 2020. Each line represents a mean daily ground temperature profile averaged over the five locations where data loggers where installed in 2, 10, and 50 cm depth. (c) Mean multiannual (black) and multiseasonal (Bega: green, Belg: blue, Kiremt: purple) ground temperature gradients in 2 cm depth between 3493 and 4377 m considering all data (excluding the warm-biased GT16 logger from a southern slope and the cold-biased GT17 logger from a northern slope) from February 2017 to January 2020.

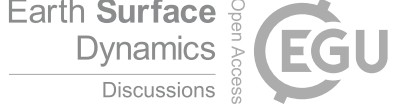

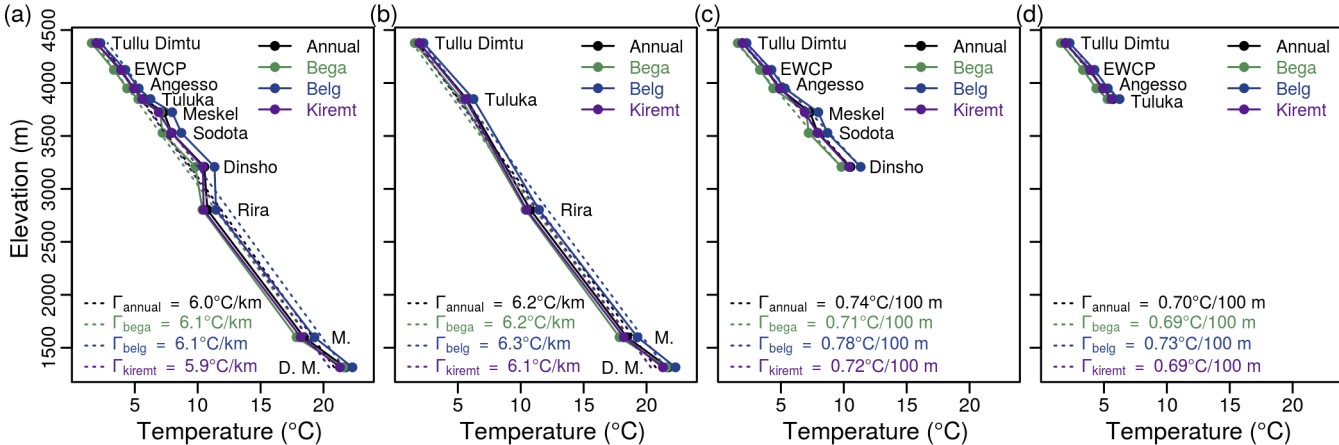

**Figure 9.** Mean multiannual and multiseasonal (Bega: Nov/Dec/Jan/Feb, Belg: Mar/Apr/May/Jun, Kiremt: Jul/Aug/Sep/Oct) lapse rate of air temperature (a) in the Bale Mountains, (b) along the Harenna Escarpment, (c) along the northern declivity, and (d) on the Sanetti Plateau between February 2017 and January 2020. The lapse rate for the northern declivity and Sanetti Plateau is given in °C per 100 m because of the relatively small elevation range (ca. 1200 and 500 m).

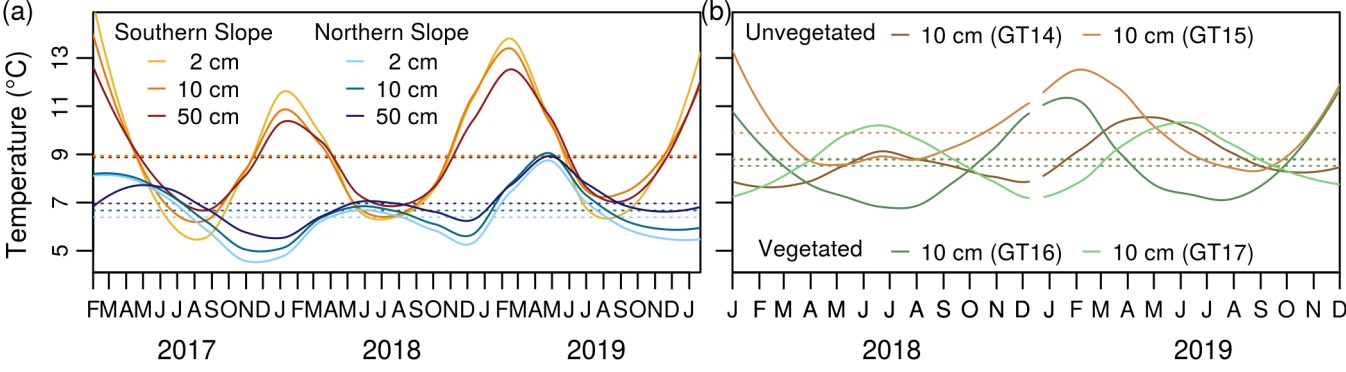

**Figure 10.** (a) Comparison of the seasonal ground temperature variations in 2, 10, and 50 cm depth between the northern and southern slope of Mount Wasama. Data from the southern slope at 4153 m are from the loggers GT16/GT02/GT03, and data from the northern slope at 4181 m are from the loggers GT17/GT05/GT06. (b) Comparison of seasonal ground temperature variations in 2, 10, and 50 cm depth between locations with and without Erica at the slopes of Tullu Dimtu. A local regression with a smoothing span of 0.32 was applied to derive seasonal ground temperature variations from hourly measurements.

applications, are suitable for short-term (< 1 year) ground temperature measurements and experiments at high elevations. Both loggers measured nearly the same mean ground temperature (8.46 vs. 8.48 °C). Only the standard deviation of the low-cost logger was a bit larger (9.1 vs. 7.3 °C) since it was installed minimal closer to the surface.



### 4.3 Modelled palaeo ground temperatures

At the three locations on the Sanetti Plateau (Tullu Dimtu, EWCP Station, and Tuluka), where ground temperatures and a set of meteorological variables were measured concurrently, the ground temperature is mainly controlled by air temperature and global radiation (Fig. 11). The two variables can explain together about 75 ± 3 % of the ground temperature variance

(Table 4). Ground temperature and the other meteorological variables do not show any significant linear relationship what is not surprising in view of the non-consideration of ground moisture. Precipitation, relative humidity, and wind speed affect ground moisture as well as evaporation and therefore alter the energy balance at the surface and energy transfer into the ground. Measuring and considering ground moisture (directly at the AWS) would likely help to reduce the uncertainties of the applied multiple linear regression models (RMSE of 3-4 °C). Nevertheless, the established relationship between ground temperature

and air temperature/global radiation is strong enough to use the models for a first palaeoclimatic reconstruction experiment. To obtain mean annual ground temperatures associated with deep seasonal ground frost or sporadic permafrost ($T_{ground}$ < -1 °C) at the elevation of the lowermost stone stripes on the southern Sanetti Plateau during their time of formation, the three tested linear models require a mean ground temperature depression of ~12 °C. This would translate into a mean air temperature depression of 7.6 ± 1.3 °C (the error is the standard deviation of the three model outputs) which would imply a mean annual

air temperature on the southern plateau of -1.9 ± 1.3 °C. The deduced stronger decrease of the ground temperature over the air temperature is due to the observed modern relationship. A cooling/warming of the air of 1 °C relates to a decrease/increase of the ground of 1.6-1.9 °C and vice versa (see Table 4 and Fig. 11). The geophysical reasons for this statistical relationship can be manifold. They are associated with the radiative forcing and energy exchange between the atmosphere and ground, which in turn is affected by many factors ranging from insolation, air pressure, relative humidity to the thermal conductivity, specific

heat capacity, density, humidity, albedo, etc. of the ground.
Over a year, the seasonal mean daily ground temperature fluctuations between the surface and 50 cm depth were theoretically large enough to freeze and thaw the upper half metre of the ground (Fig. 12). However, seasonal freezing and thawing below 50 cm depth seems unlikely in the past if the ground properties and seasonal temperature fluctuations were similar like today. A mean annual ground temperature in the order of -1 °C seems critical for the formation of deep seasonal frost on the Sanetti

Plateau since much warmer temperatures would prevent seasonal freezing and lower temperatures seasonal thawing.

### 5 Discussion

Comprehensive geomorphological investigations in combination with different field measurements, [36]Cl surface exposure dating, and statistical ground temperature modelling presented here provide novel insights in the distribution, characteristics, and palaeoclimatic implications of modern and relict periglacial landforms in the tropical Bale Mountains in the southern Ethiopian

Highlands. Modern diurnal and seasonal frost phenomena like small-scale patterned grounds, solifluction lobes, frozen waterfalls, and needle ice are limited to the Sanetti Plateau, the highest peaks, and the upper part of the northern valleys above 3900 m. Relict periglacial landforms are more abundant and much more pronounced in the Bale Mountains. Besides extensive blockfields along the southern and western Harenna Escarpment, large sorted stone stripes on the southern and western



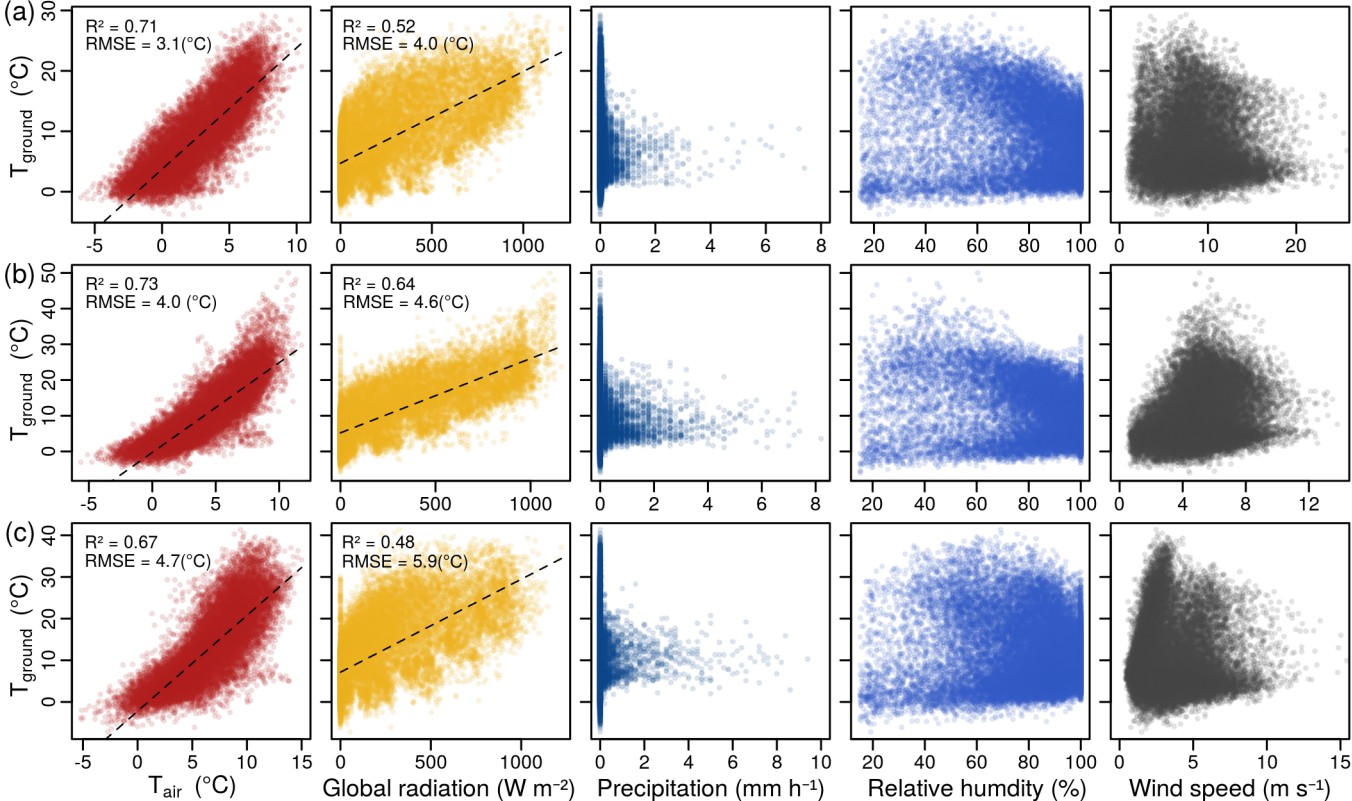

**Figure 11.** Relationship between hourly ground temperatures in 2 cm depth and different meteorological variables at three different locations: (a) Tullu Dimtu (GT13 vs. BALE001), (b) EWCP Station (TM04 vs. BALE009), (c) Tuluka (GT07 vs. BALE002).

Sanetti Plateau between 3850 to 4150 m are the most prominent geomorphological features. These features are associated with seasonal freezing and thawing of the upper half metre of the ground. Ground temperature measurements at sixteen locations between 3493 to 4377 m over a three-year period (2017-2020) verify seasonal temperature variations and frequent nocturnal frost on the plateau and in the valleys. However, the frost penetrates only the uppermost centimetres of the ground. The mean

5  annual present-day ground temperature between the southern stone stripes (~11 °C) is way off from permafrost conditions. Experimental modelling suggests that a distinct ground temperature depression of ~12 °C and air temperature depression of 7.6 ± 1.3 °C would be necessary for the formation of deep seasonal frost at the elevation of the stone stripes. The main aim of the following discussion is to elaborate when and how the large structures may have formed and what their presence implies for the palaeoclimate and palaeogeoecology of the tropical Ethiopian Highlands.

10  Patterned grounds comprising sorted stone stripes, circles, and polygons are a common feature of periglacial environments and are known from the Arctic (e.g. Nicholson, 1976; Hallet, 2013), Antarctic (e.g. Hallet et al., 2011), mid latitudes (e.g. Richmond, 1949; Miller et al., 1954; Ball and Goodier, 1968; André et al., 2008), and high mountains worldwide (e.g. Francou et al., 2001; Matsuoka, 2005; Bertran et al., 2010). They have also been detected on other celestial bodies like Mars (e.g. Man-

Earth **Surface**
**Dynamics**
Discussions

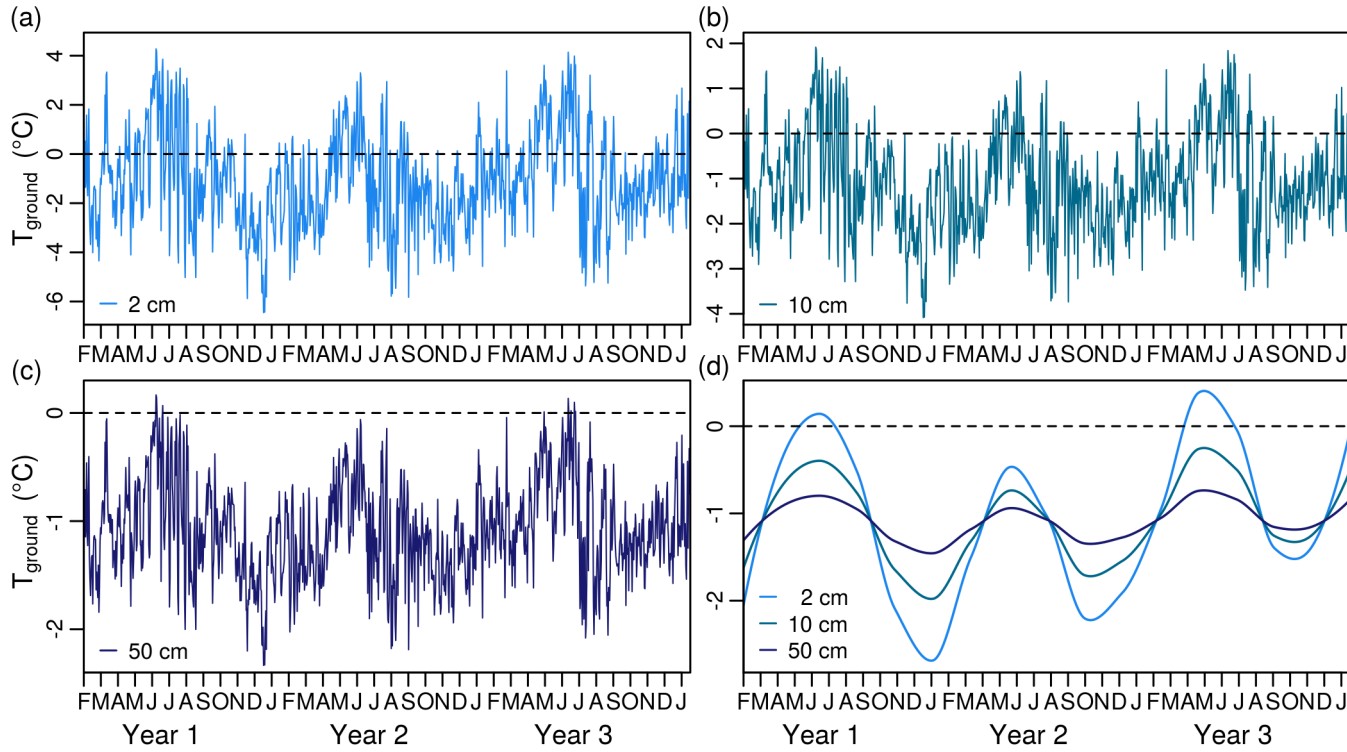

**Figure 12.** Simulated daily mean ground temperatures in (a) 2 cm, (b) 10 cm, and (c) 50 cm depth on the southern Sanetti Plateau (3877 m) assuming a decrease in temperature of 7.6 ± 1.3 °C and decrease in global radiation of 30 W/m² relative to present-day conditions. (d) A local regression with a smoothing span of 0.32 was applied to derive seasonal ground temperature variations from daily mean values.

gold, 2005; Balme et al., 2009). Small-scale sorted stone stripes in the order of centimetres to decimetres, which are associated with superficial diurnal freeze-thaw cycles, are typical for several mid-latitude and also high tropical mountains (e.g. Francou et al., 2001; Matsuoka, 2005). However, large sorted stone stripes comparable to those on the Sanetti Plateau (10-15 m wide and 100-1000 m long) have not been reported from any other tropical mountain and seem to be a rare phenomenon in general.

5 In contrast to small-scale features which do not necessarily require mean annual air temperatures below freezing, large sorted patterned grounds like stone circles and polygons that are well-documented for the High Arctic (e.g. Kessler and Werner, 2003; Hallet, 2013) occur commonly in permafrost areas with mean annual air temperatures of -4 to -6 °C (Goldthwait, 1976). The only other location worldwide where stone stripes in the same order of magnitude as on the Sanetti Plateau and even larger have been described are the non-volcanic Falkland Islands in the South Atlantic (André et al., 2008).

10 The vernacular term for extensive blockstreams and stone stripes in the Falkland Islands is "stone runs". Stone runs cover large parts of the eastern and western island and are linked to quartzite outcrops in the elevated areas (50-700 m). The stone stripes in the Falkland Island show some interesting similarities and differences with the features on the Sanetti Plateau. They occur in clusters at gentle slopes (inclination: 1-10°), are several hundred meters long, several meters wide, consist of large





angular blocks (up to 2 m wide and 5 m long), and originate in some cases from eroded ridges and summit areas. As on the Sanetti Plateau, the coarse stone stripes in the Falkland Islands run parallel downslope and alternate with stripes of fine-grained material of similar width (André et al., 2008). However, the partial emergence of stone stripes from blockfields and downslope transition into vast blockstreams as it is typical for the Falkdland Islands is uncommon for the Bale Mountains where

the stripes are restricted to the plateau and the blockfields to the southern and western escarpment. Also the geological (volcanic vs. sedimentary and metamorphic rocks), climatic (continental vs. oceanic), and geographical setting (tropical mountain vs. mid-latitude island) between the Bale Mountains and Falkland Islands differs considerably. A link between both locations is the coexistence of coarse and fine-grained material (large angular blocks and regolith) and the evidence for glaciations and colder conditions during the Pleistocene (Clapperton, 1971; Clapperton and Sudgen, 1976; Groos et al., in revision).

The origin and genesis of the stone runs in the Falkland Islands has been discussed controversially over the last one hundred years and numerous theories have been proposed to explain their formation as a result of different interconnected periglacial processes (frost shattering, frost heave, frost sorting, etc.). Based on a literature review and micromorphological analyses, André et al. (2008) come to a more nuanced conclusion and consider the stone runs as complex polygenetic landform. The authors hypothesise that the parent material (blocks and regolith) formed under subtropical or temperate conditions during

the Neogene/Palaeogene and interpret the stone runs as the product of subsequent frost-sorting during the cold stages of the Pleistocene. Nevertheless, the understanding of the physical processes underlying the frost-related sorting of such large blocks is still fragmentary (Aldiss and Edwards, 1999). The limited process understanding, the small number of analogies worldwide, and the lack of a cross-section profile complicate the interpretation of the stone stripes on the Sanetti Plateau in the Bale Mountains, but the following observations and findings suggest that deep seasonal frost and periglacial processes played a major role

in the formation of this landform.

As aforementionend, the Bale Mountains were covered by an extensive ice cap and experienced a pronounced cooling of 4-6 °C between 50-30 ka (Ossendorf et al., 2019; Groos et al., in revision). It is important to note that the stone stripes on the western and southern Sanetti Plateau are located beyond the glacial remains and the assumed maximum extent of the former ice cap. The obtained $^{36}$Cl surface exposure ages of >600 ka ($^{36}$Cl has reached saturation in two samples) for the western and

of 84 ± 4, 281 ± 12, and 281 ± 13 ka for the southern stone stripes predate the local last glacial maximum. However, the exposure ages probably do not represent the timing of formation or stagnation of the features. Since the sampled trachytic blocks and columnar basalt originate from eroded cliffs and volcanic plugs, it is likely that they were exposed to cosmic radiation during and prior to the formation of the stone stripes. The well-preserved morphology, the absence of erratic boulders in the surrounding area, and the high $^{36}$Cl concentrations indicate that the stone stripes have not been eroded and deformed by a

dynamic and warm-based glacier. However, we cannot compeletely rule out that stagnant, cold-based, and relatively shallow ice covered the stripes for several thousand years. Such an ice cover would not have been sufficient to zero the inheritence (high $^{36}$Cl concentrations). We interpret the hardly weathered surface of the angular blocks and little reoccupation by vegetation as an indication for a younger formation stage of the stone stripes coinciding probably with the coldest and driest phase in Africa (30-15 ka) during the last glacial period (e.g. Tierney et al., 2008). The ice extent in the Bale Mountains at ~18 ka was slightly

smaller compared to the local LGM despite the general cooling trend in Eastern Africa. The lack of any evidence for a major





glacier advance between the local LGM and the ~18 ka stage might be indicative of a cold and dry climate which provides ideal conditions for periglacial processes.

A precondition for the formation of patterned grounds is cyclic freezing and thawing of a decimetre- to meter-thick layer and the coexistence of fine-grained material and larger stones or blocks (Kessler et al., 2001; Kessler and Werner, 2003). Both

large blocks and fine-grained material are present on the Sanetti Plateau. A decimetre-thick regolith layer rich in silt and loam covers the underlying bedrock of the plateau (Lemma et al., 2019). Whether the regolith has developed during the Pleistocene or during warmer periods before, as suggested for the Falkland Islands, remains unclear. The accumulation of trachytic blocks and columnar basalt at some places on the plateau is probably related to intensive frost wedging at cliffs and volcanic plugs during the last glacial cycle. As our ground temperature modelling suggests, a half-meter-thick frozen layer could have formed

seasonally on the plateau outside the glaciated area as the result of a pronounced air temperature cooling of 7.6 ± 1.3 °C during the coldest period of the last glacial cycle. Deep frozen grounds and sporadic permafrost still exist at some of the highest tropical and subtropical mountains in Africa (Kaser et al., 2004; Vieira et al., 2017). Potential evidence for past sporadic permafrost in the Bale Mountains exists in the northeastern Togona Valley, which was covered by a 8 km long valley glacier during the Late Pleistocene. During or after deglaciation of the lower part of the valley, two large landslides (0.5 and 1.5 km long; see

Fig. 5) occurred between the ~18 ka and ~15 ka moraine stages and might have been triggered by slope destabilisation due to thawing permafrost.

The strongest arguments for the stone stripes being the "final" product of frost heave and sorting is their configuration as well as the presence of less-pronounced relict large sorted stone polygons in the highest flat parts of the western plateau. The width of the alternating fine-grained and coarse stone stripes (about 10-20 times larger than the average block size), the absence of

larger blocks in the fine-grained stripes, the axis orientation of the stripes parallel to the greatest slope, and the convergence of individual narrower branches to wider single stripes is remarkably similar to patterned grounds at gentle slopes predicted by numerical models after several hundred freeze-thaw cycles (Werner and Hallet, 1993; Mulheran, 1994; Kessler et al., 2001; Kessler and Werner, 2003). Such numerical models can reproduce the self-organization of different sorted grounds by varying just a few parameters (mainly stone concentration, hillslope, and degree of lateral confinement) and need about 500 to 5000

freeze-thaw cycles to form similar stripe patterns as found on the southern Sanetti Plateau (Fig. 5). Less cycles would lead to a more random configuration and more cycles would eliminate the smaller branches and lead to a "perfect" sorting of the stripes (see Fig. 2 in Werner and Hallet, 1993). Assuming downslope displacement rates of 10-50 cm per year (or cycle) for clasts as it is observed for small-scale periglacial features in the tropics (Francou and Bertran, 1997) would require a similar number of cycles (about 400 to 2000) to form the 200 m long stone stripes on the southern plateau. In view of the length of the coldest

phase of the last glacial period, the formation of the stone stripes on the plateau in proximity to the ice cap over several hundred to thousand cycles/years is a plausible scenario.

Instead of seasonal variations, longer freeze-thaw cycles could theoretically also explain the formation of the stone stripes on the Sanetti Plateau. A formation over several cold stages during the Pleistocene as proposed for the stone runs in the Falkland Islands (Wilson et al., 2008) is conceivable. This would imply the formation of sporadic permafrost during the colder periods

and complete thawing of the ground during the warmer periods of the Pleistocene. The "old" exposure ages would generally





support such a scenario, although the mismatch between the high $^{36}$Cl concentrations of the western stone stripe compared to the lower concentrations of the southern stone stripe would remain an open question. However, due to the well-preserved morphology of the stone stripes, the verified seasonal ground temperature variations on the Sanetti Plateau, and the absence of further evidence for the formation over several cold stages, we propose seasonal freezing and thawing during the last glacial

cycle as the main mechanism for the formation of the stripes.

The presence of sorted stone stripes and other relict frost-related landforms on the Sanetti Plateau and along the Harenna Escarpment provide further evidence that the Bale Mountains underwent severe climatic and environmental changes during the Pleistocene. Both glacial and periglacial processes played a major role in shaping the afro-alpine landscape. The inferred cooling of 7.6 ± 1.3 °C needed for the formation of large patterned grounds on the Sanetti Plateau is much larger than the

temperature decrease in the Bale Mountains of 5.1 ± 0.7 °C derived from the estimated snow line depression during the local last glacial maximum 50-30 ka (Groos et al., in revision). This discrepancy might indicate a further cooling in the southern Ethiopian Highlands from the time of the local maximum glacier expansion to the global LGM (22 ± 4 ka) as seen in other climate records from Eastern Africa and worldwide (e.g. Jouzel et al., 2007; Tierney et al., 2008). Moreover, the reconstructed temperature decrease from the Ethiopian Highlands in comparison with the reconstructions from the lower-situated Congo

Basin and Lake Tanganyika in the order of 4 to 4.5 °C (Weijers et al., 2007; Tierney et al., 2008) reveals an amplified tropical cooling at high elevations in Eastern Africa during the global LGM. Strong evidence for such a cooling is also provided by sea surface and air temperature reconstructions from different lakes along an elevational transect in Eastern Africa. Loomis et al. (2017) explain the observed elevational trend with a steeper tropical lapse rate ($\Gamma_{LGM}$ = 6.7 ± 0.3 °C km$^{-1}$ vs. $\Gamma_{modern}$ = 5.8 ± 0.1 °C km$^{-1}$) related to a drier atmosphere during the global LGM. The present-day mean annual lapse rate and ground temperature

gradient on the Sanetti Plateau of 0.7 °C per 100 m (Fig. 8 and 9) is larger than the palaeo ($\Gamma_{LGM}$) and modern ($\Gamma_{modern}$) East African lapse rate between 474 and 3081 m. This emphasises that attention should be drawn to temporal as well as vertical changes in the lapse rate when reconstructing or simulating the palaeoclimate of the tropics. Despite improvements, global climate models still tend to underestimate the cooling at high elevations in the tropics during the last glacial cycle (Loomis et al., 2017).

The palaeoclimatic and environmental findings presented here have direct implications for the settlement history and ecology of the Bale Mountains. Latest archaeological excavations at 3469 m in the northwestern part of the Bale Mountains along with biogeochemical, zoogeographical, and glacial chronological investigations reveal that Middle Stone Age foragers resided in the highlands already between 47 to 31 ka and made use of the available alpine resources (Ossendorf et al., 2019). Why the residential site was abandoned after 31 ka is unclear, but might be related to a gradual cooling and dessication of the highlands

until the global LGM as suggested by the large-scale periglacial landforms and lack of major glacier advances between 50-30 and ~18 ka (Groos et al., in revision). Moreover, the temperature depression, ice cover, and periglacial conditions must have also affected the habitat of endemic mammal species like the Ethiopian wolf, giant mole-rat, and mountain nyala that currently populate the Sanetti Plateau and upper valleys (Miehe and Miehe, 1994). Due to the absence of any evidence for glacial extinction events in the region, we conclude that endemic plants and mammals as well as Middle Stone Age foragers coped with

the harsh climatic and environmental conditions in the Ethiopian Highlands during the Pleistocene.



Many questions regarding the relict periglacial processes and landforms in the Bale Mountains remain open due to the pioneering and experimental character of this study and may hopefully stimulate further research on this topic. A key challenge for better understanding the landscape evolution on the Sanetti Plateau is the development of a robust geochronology. The age of the volcanic plugs, the formation time of the regolith and stone stripes as well as the deglaciation history of the former ice cap

on Tullu Dimtu are uncertain. Moreover, information on the depth and internal structure (grain size distribution, mineral composition, etc.) of the coarse and fine-grained stone stripes would provide additional insights into the genesis of the landforms. To reduce the uncertainty of the statistical model applied for ground temperature simulations and air temperature reconstructions, considering the impact of moisture on the thermal conductivity and heat capacity of the ground as well as energy fluxes into the ground is necessary. It is possible that ground moisture and stagnant water played a more important role during the

Late Pleistocene than today due to (perma)frost-induced waterlogging and perennial melting of snow and ice. Why the relict patterned grounds are restricted to the southern and western Sanetti Plateau can be explained by the assumed preconditions for their formation: a relatively flat and unglaciated terrain, presence of coarse and fine grained material, deep ground frost, and absence of a thick snow layer.

## 6   Conclusions

This contribution provides further evidence that the tropical Bale Mountains in the southern Ethiopian Highlands were subject to severe climatic and environmental changes during the Late Pleistocene. Both glacial and periglacial processes have shaped the afro-alpine environment. Compared to the modern nocturnal and seasonal frost phenomena, relict periglacial landforms like blockfields along the Harenna Escarpment and sorted stone stripes on the Sanetti Plateau are much larger and more developed. The large sorted stone stripes are exceptional for the tropics and probably formed under periglacial conditions in proximity

of the palaeo ice cap on Tullu Dimtu during the coldest period(s) of the last glacial cycle. We hypothesise that the slightly inclined and unglaciated areas of the Sanetti Plateau, the coexistence of regolith and large blocks, the occurrence of deep seasonal frost, as well as relatively dry conditions beyond the ice cap provided ideal conditions for frost heave and sorting and the formation of large patterned grounds. Based on our ground temperature measurements and modelling experiment, we propose a distinct ground temperature depression of ~12 °C and air temperature depression of 7.6 ± 1.3 °C as precondition for

the formation of deep ground frost on the Sanetti Plateau. The novel idea of using a statistical relationship between measured present-day ground temperatures and meteorological variables to assess past climatic changes through the simulation of palaeo ground temperatures has also potential for other regions where relict frost-related periglacial landforms exist. Comparing the reconstructed air temperature depression from the Bale Mountains with climate records from lower elevations in Eastern Africa emphasises a strongly amplified cooling at high elevations that has already been outlined for many other tropical mountains.

Such a cooling in tandem with the extensive glaciation and frost action must have dramatically affected the habitat of endemic mammal species like the Ethiopian wolf, giant mole-rat, and mountain nyala that currently populate the Sanetti Plateau and upper valleys of the Bale Mountains. Attention should therefore be given to the amplified middle troposphere cooling when reconstructing and modelling climatic and geoecological changes in the tropical mountains of Eastern Africa.



*Data availability.* The ground temperature data (Tables S1-4) can be downloaded from the open access library PANGAEA (provisional link until data are published: https://filesender.switch.ch/filesender/?vid=06bf00d3-d750-e748-82f8-00006e12ef78). Raw data, aerial images, additional field photos, etc. are available upon request by email to the corresponding author.

*Author contributions.* ARG, NA, and HV designed the research concept, conducted the geomorphological mapping, sampled the stone
stripes for exposure dating, and installed the ground temperature loggers. ARG and NA processed the rock samples in the laboratory. FH set up the weather stations. LW conducted the GPR measurements and serviced the weather stations. FH, LW, and TN processed and provided the meteorological data. ARG and JN processed the ground temperature data, conducted the statistical analysis, and performed the ground temperature simulations. ARG drafted the manuscript and figures with contributions from all authors.

*Competing interests.* The authors declare that they have no conflict of interest.

*Acknowledgements.* This research was funded by the Swiss National Science Foundation (SNF, grant no. 200021E-165446/1) and the German Research Foundation (DFG) in the framework of the joint Ethio-European DFG Research Unit 2358 "The Mountain Exile Hypothesis". We thank the Ethiopian Wildlife Conservation Authority, the College of Natural and Computational Sciences (Addis Ababa University), the Department of Plant Biology and Biodiversity Management (Addis Ababa University), the Philipps University Marburg, the Frankfurt Zoological Society, the Ethiopian Wolf Project, and the Bale Mountains National Park for their cooperation and kind permission to conduct
field work. We are grateful to Mekbib Fekadu, Wege Abebe, Katinka Thielsen, Tiziana Koch, Aschalew Gashaw, Terefe Endale, Geremew Mebratu, Beriso Kemal, Mohammed Kedir, Edris Abduku, Sabrina Erlwein, Lukas Munz, Julian Struck, and Bruk Lemma for contributing to the preparation and implementation of the field work, Serdar Yesilyurt for support in the lab, and Armin Rist for the fruitful discussion. Special thanks also go to the DigitalGlobe Foundation for providing high-resolution WorldView-1 satellite images of the Bale Mountains (granted to ARG) and to the developers/maintainers of the free and open-source software used in this study (R, QGIS, OpenDroneMap,
MeshLab, LibreOffice, etc.).





# Appendix A: Catalogue of periglacial features

**Table A1.** Overview of periglacial landforms and other characteristic geomorphological features in the Bale Mountains mapped in the field and on satellite images. A compilation of glacial landforms in the Bale Mountains is provided by Groos et al. (in revision).

| ID | Landform / Feature | Status | Latitude (°N) | Longitude (°E) | Elevation (m) | Slope (°) | Aspect (°) |
|----|--------------------|--------|---------------|----------------|---------------|-----------|------------|
| 1  | Frozen waterfalls  | active | 6.91508 | 39.76298 | 3980 – 4000 | 32 – 37 | 350 – 10 |
| 2  | Needle ice         | active | 6.91784 | 39.76978 | 3925 – 3935 | 0 | |
| 3  | Sorted stone nets  | active | 6.84253 | 39.77714 | 4110 – 4140 | 0 | |
| 4  | Scree slope        | active | 6.92509 | 39.78395 | 3930 – 4090 | 18 – 37 | 110 – 120 |
| 5  | Solifluction lobes | active | 6.92699 | 39.77194 | 4130 – 4190 | 20 – 22 | 150 – 170 |
| 6  | Sorted stone stripes  | relict | 6.78692 | 39.79278 | 3865 – 3880 | 3 – 9 | 290 – 70 |
| 7  | Sorted stone stripes  | relict | 6.79496 | 39.81503 | 3880 – 3940 | 3 – 7 | 70 – 180 |
| 8  | Sorted stone stripes  | relict | 6.85486 | 39.72071 | 4020 – 4100 | 2 – 9 | 330 – 350 |
| 9  | Sorted stone stripes  | relict | 6.85336 | 39.71750 | 4020 – 4140 | 2 – 9 | 330 – 350 |
| 10 | Sorted stone stripes  | relict | 6.85432 | 39.71263 | 4000 – 4070 | 2 – 9 | 330 – 350 |
| 11 | Sorted stone stripes  | relict | 6.85264 | 39.70884 | 3940 – 4100 | 2 – 9 | 330 – 350 |
| 12 | Sorted stone stripes  | relict | 6.91414 | 39.60676 | 3715 – 3730 | 2 – 9 | 270 – 290 |
| 13 | Sorted stone polygons | relict | 6.83843 | 39.70631 | 4000 – 4100 | 0 – 4 | 180 – 200 |
| 14 | Sorted stone polygons | relict | 6.84533 | 39.71969 | 4120 – 4170 | 0 – 4 | 330 – 350 |
| 15 | Blockfield         | relict | 6.76713 | 39.78794 | 3690 – 3800 | 19 – 25 | 240 – 250 |
| 16 | Blockfield         | relict | 6.82818 | 39.78168 | 3970 – 4030 | 12 – 15 | 260 – 270 |
| 17 | Blockfield         | relict | 6.83016 | 39.71949 | 3700 – 3940 | 17 – 19 | 200 – 220 |
| 18 | Blockfield         | relict | 6.84541 | 39.69772 | 3800 – 3880 | 9 – 11 | 300 – 310 |
| 19 | Blockfield         | relict | 6.85245 | 39.69704 | 3700 – 3830 | 12 – 14 | 260 – 270 |
| 20 | Blockfield         | relict | 6.86119 | 39.69388 | 3550 – 3820 | 20 – 24 | 250 – 270 |
| 21 | Blockfield         | relict | 6.86848 | 39.69701 | 3600 – 3880 | 20 – 24 | 320 – 330 |
| 22 | Scree slope        | relict | 6.89194 | 39.89919 | 3890 – 3940 | 20 – 23 | 300 – 320 |
| 23 | Scree slope        | relict | 6.88617 | 39.89236 | 3930 – 3980 | 20 – 26 | 350 – 360 |
| 24 | Scree slope        | relict | 6.91829 | 39.77699 | 4070 – 4110 | 24 – 25 | 350 – 360 |
| 25 | Scree slope        | relict | 6.95343 | 39.76925 | 4045 – 4065 | 24 – 25 | 290 – 310 |
| 26 | Scree slope        | relict | 6.93937 | 39.78443 | 4080 – 4110 | 21 – 25 | 290 – 310 |
| 27 | Scree slope        | relict | 6.94363 | 39.78672 | 4055 – 4100 | 23 – 25 | 350 – 360 |
| 28 | Scree slope        | relict | 6.94764 | 39.79058 | 4080 – 4150 | 24 – 27 | 10 – 20 |
| 29 | Landslide          | relict | 6.92268 | 39.89833 | 3490 – 3720 | 2 – 30 | 60 – 70 |
| 30 | Landslide          | relict | 6.92644 | 39.90251 | 3490 – 3650 | 2 – 40 | 160 – 170 |





## Appendix B: GPR system settings

**Table B1.** System settings of the used Pulse EKKO PRO GPR.

| Setting type | Setting | Setting type | Setting | Setting type | Setting |
|---|---|---|---|---|---|
| Frequency: | 1000 MHz | Survey type: | Reflection | Start offset: | 0 m |
| Time window: | 30 ns (1.6 m) | Step size: | 0.010 m | GPR trigger: | Odometer |
| Sampling Interval: | Normal (100 ps) | Calibration: | 1080.0 | Antenna separation: | 0.15 m |
| Stacks: | 4 | Transmitter: | pE Pro Auto | Antenna polarization: | broadside |
| Velocity: | 0.12 m ns$^{-1}$ | Receiver: | pulseEKKO Pro | Antenna orientation: | Perpendicular |





## Appendix C: Cosmogenic $^{36}$Cl data

**Table C1.** Major and trace element data of the six rock samples from the Sanetti Plateau.

| Rock sample | O * | C * | Na * | Mg * | Al * | Si * | P * | K * | Ca * | Ti * | Mn * | Fe * | B ** | Sm ** | Gd ** | U ** | Th ** |
|---|---|---|---|---|---|---|---|---|---|---|---|---|---|---|---|---|---|
| BS01 | 57.88 | 5.13 | 1.74 | 5.61 | 7.40 | 21.97 | 0.09 | 0.62 | 7.86 | 1.44 | 0.15 | 9.10 | 3 | 3.3 | 3.6 | 0.3 | 1.1 |
| BS02 | 56.64 | 5.09 | 1.70 | 5.50 | 7.23 | 21.13 | 0.14 | 0.61 | 7.90 | 1.41 | 0.15 | 9.13 | 11 | 3.8 | 4.1 | 0.3 | 1.3 |
| BS03 | 56.17 | 4.98 | 1.68 | 5.23 | 7.52 | 20.90 | 0.15 | 0.60 | 8.00 | 1.44 | 0.14 | 8.97 | 12 | 3.8 | 4.1 | 0.3 | 1.2 |
| BS04 | 54.79 | 3.85 | 2.50 | 3.56 | 8.30 | 22.86 | 0.14 | 0.81 | 6.96 | 1.42 | 0.15 | 8.81 | 6 | 4.3 | 4.4 | 0.4 | 1.7 |
| BS05 | 47.82 | 0.68 | 5.01 | 0.21 | 9.09 | 28.42 | 0.05 | 3.59 | 1.92 | 0.24 | 0.19 | 4.39 | 1 | 6.4 | 4.8 | 2.9 | 14.8 |
| BS06 | 46.47 | 0.66 | 5.16 | 0.18 | 9.42 | 26.99 | 0.05 | 3.64 | 1.90 | 0.24 | 0.19 | 4.41 | 15 | 6.7 | 4.9 | 1.9 | 15.5 |

Data from Groos et al. (in revision). *Unit = % w/w. **Unit = ppm.

**Table C2.** Cosmogenic $^{36}$Cl data and surface exposure ages of the rock samples from the Sanetti Plateau.

| Rock sample | Rock dissolved (g) | $^{35}$Cl Spike (mg) | Cl (ppm) | $^{36}$Cl ($10^5$ At g$^{-1}$) | Exposure age (ka)* | Exposure age (ka)** | Exposure age (ka)*** |
|---|---|---|---|---|---|---|---|
| BS01 | 30.0307 | 2.5682 | 20.7 ± 0.08 | 30.44 ± 0.82 | 84.2 ± 3.7 | 86.7 ± 3.9 | 90.9 ± 4.3 |
| BS02 | 30.0068 | 2.5584 | 31.5 ± 0.07 | 85.93 ± 1.63 | 280.8 ± 11.8 | 332.0 ± 16.5 | 781.3 ± 40.0 |
| BS03 | 29.9887 | 2.5584 | 29.1 ± 0.04 | 85.66 ± 2.45 | 280.8 ± 13.0 | 334.5 ± 17.6 | 859.4 ± 46.2 |
| BS04 | 29.9982 | 2.5652 | 40.9 ± 0.22 | 153.56 ± 2.58 | 620.1 ± 28.0 | | |
| BS05 | 30.0349 | 2.5719 | 1027.6 ± 11.19 | 1268.53 ± 25.03 | - | - | - |
| BS06 | 30.0705 | 2.5682 | 1228.0 ± 13.43 | 1394.82 ± 46.40 | - | - | - |

Data from Groos et al. (in revision). *Erosion rate = 0 mm ka$^{-1}$. **Erosion rate = 1 mm ka$^{-1}$. ***Erosion rate = 2 mm ka$^{-1}$



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
