# Peer review of "The enigma of relict large sorted stone stripes in the tropical Ethiopian Highlands"

_Earth Surface Dynamics, 2020_

## Referee Comment (RC1) · Stefan Grab (Referee) · 8 Jul 2020

Referee report on: Implications of present ground temperatures and relict stone stripes in the Ethiopian Highlands for the palaeoclimate of the tropics Authors: Groos et al.

General comments I am pleased to see that the Bale Mnts are receiving this geomorphic attention it deserves. I have worked on a variety of periglacial landforms in various African environments over many years now - but to me the large scale sorted stone stripes presented in this paper from the Sanetti Plateau, are the most special and unique periglacial landforms I have yet seen in Africa. They are truly special and must rank amongst the largest and best examples globally, I would think. So work on

these is certainly called for and important to publish. I think the big challenge with these amazing landforms is ascertaining when they developed, how long they were actively forming for and when they might have become inactive (relict) periglacial phenomena. The second great challenge is ascertaining how they formed, because any misunderstanding as to their formation has serious implications to any climatic controls we attach to their genesis. I think these challenges are very real for this paper and when I read the work I see that this is where the paper has its struggles. I have several major concerns with this paper, which I will outline next, but at the same time wish to also assist with providing suggestions that might help rework this paper into something that might be publishable.

Major concerns to address 1. The paper is far too long and tries to tackle too many things with too much detail, such that the connections between the various bits of collected data/information, become somewhat muddled and lost in the discussion/conclusion. The extent of detail to such things as instrumentation and story behind the logger battery issues etc may be valuable to place in a technical report or PhD thesis, but is not suitable for a journal publication. The text requires substantial trimming down and tightening up throughout. 2. Although the written style is relatively uncomplicated and for the most part satisfactory, there is a tendency towards colloquial language style, which is not suitable. The written style thus requires considerable improvement for publication. 3. The Scientific methods are notable and impressive for such a region given the logistical hurdles. However, great or large quantities of data may not always be the most useful or necessary data for the study objectives. - While it is great having 36Cl results for the landforms, these raise more questions than provide answers. These do not necessarily inform us when the landforms first developed, or how long they were actively forming for, or when they became 'periglacially inactive'. So, despite all the efforts in obtaining exposure ages, the authors are still left with merely assuming that the landforms are of Late Glacial age. Such an assumption might be reached without the exposure ages and have in fact also been made for other openwork block deposits (e.g. block streams) in the high Drakensberg (southern
Africa) – see for e.g. Boelhouwers et al. (2002). Without any real sense of timeframe, it is impossible to use the landforms for any palaeo-climate reconstructions. - Great effort was also undertaken with the Ground-penetrating radar measurements; something not previously done for periglacial landforms in Africa. However, the results to me do not show much that is of significance - and so does not add enough value to provide for anything noteworthy to add to the discussion or meet the aim/objectives of this paper. I would like to be proven wrong here – so if the authors can indeed use these data in a way that enhances/strengthens the discussion, then that would be good. - The authors provide considerable temperature data (ground and air). In fact I think too much is attempted with these temperature records and in the process of trying too much with it (also too many graphs), the scientific value and merit is lost. I will elaborate on temperature data separately as this constitutes a major concern. 4. Temperature data: While the temperature data recorded at various localities might be used for various scientific purposes, I think the way in which the data have been used in this paper requires very careful reconsideration. - The work is built upon the presumption that the sorted stripes are a product of past seasonal or sporadic permafrost that would have required ground temps of -1°C . . .or a thermal reduction of around 12°C from those recorded more recently. And the authors argue on their modelling basis that air temps would thus have been lowered by around 7.6°C. This is all highly presumptuous and very controversial. In the first instance, ground temperatures were measured in the finer textured soil stripes and not within the openwork block deposits. When these features first formed, they may have formed in a scree of such open-work block deposition because of unique localized air flow (cooling) with depth through such openwork material, thus possibly creating 'pockets' of long lasting frozen ground phenomena (be it extended seasonal freeze or permafrost). So, the soils in which the authors have done their measurements may not have been as extensively frozen as for instance in the adjacent blocky material. Please see some published work which has shown enhanced cooling through blocky periglacial phenomena (e.g. Harris & Pedersen (1998) show much colder ground thermal conditions below blocky

materials than finer textured regolith cover). - A further point is that the authors have not considered the likely thermal impacts of snow in a palaeo-environmental context. It is thus impossible to begin modelling likely air temperature reductions unless we know 1) the actual palaeo-ground temperature at exactly the same site as the contemporary measurements were taken (which is assumed to have been below $0°C$ – but very much built on assumption as the stripes may have formed when there was deep freeze beneath the blocky material but only limited/shallow/seasonal freeze in the finer textured soil stripes), and 2) the depth, duration etc of snow which would have had an insulating effect – or maybe helped preserve cooling during particular times of the year etc. The distribution and thickness of snow across the landscape would almost certainly have had impacts on the spatial/temporal characteristics of ground freeze and thaw during past colder periods. In summary re the temperature data – it is 'stretching the data too far' to try and start modelling past air temperatures as the scientific context is far too simplistic in the way it has been presented here. In reality, the contexts are much more complicated than the authors make it out to be. At best, I think the authors can use contemporary ground temperature data to reflect on contemporary shallow soil frost phenomena.

More detailed technical matters to address: P3, line 23: How do you define 'alpine environment' . . .on what basis? Is it based on a Eurocentric view of 'alpine', or is it based on what has commonly been defined as the 'Afro-alpine' zone? I am not advocating any given view but the authors should define what they understand makes the Bale Mnts the largest African 'alpine' environment. . .as opposed to for instance the Atlas Mnts or high Drakensberg-Maloti mnt system in Lesotho, Southern Africa (in both these cases one might argue for extensive 'alpine and/or Afro-alpine' regions which are larger than that of the Bale).

P5, lines 13-14- the values of glacial extent mentioned here is according to who? Needs a reference.

P5, line 20: the authors say here that the large periglacial features are associated with

freeze-thaw processes. Unless the authors can verify that they have measured freezing and thawing dynamics here, and that these mechanisms produced these landforms, then this is a scientific assumption. So rather write as '....features are likely associated with....'

P 5, line 30 would read better to say are 'endemic to'

P13, line 11: Stone stripes apparently required a thick active layer. Why do you say it had to be thick? What do you understand to be 'thick' rather than 'thin'? What dimensions are we dealing with here? Can it be that the relict sorted stripe sorting depth might say something re to active layer thickness...or depth to which [periglacial] geomorphic mechanisms operated?

P14 – at the bottom of this page the authors list so called 'frost-induced phenomena' such as frozen waterfalls, needle ice, patterned ground and solifluction lobes. This is a bit confusing as it mixes geomorphic periglacial landform types (i.e. patterned ground and solifluction lobes) with ice types (massive ice as frozen waterfalls or needle ice developed in soil). Ground ice types might be seen as mechanistic agents, while the landforms might be seen as products of the former.

Figure 4: These are impressive photos and all valuable to add here. In photo g, I can see the patterned ground (blocky borders) – in fact they look impressive to me, but the dotted white line that the authors have placed to supposedly outline the borders (shape) do not correspond with the pattern border localities in the photo.

The caption to Figure 4 is a bit misleading I think. It informs the reader that these photos show us the 'Periglacial environment of the Bale Mnts'. In the first instance, it shows contemporary phenomena of a frozen waterfall and needle ice (i.e. the contemporary environment). These features do not qualify this to be labelled a contemporary periglacial environment as the ground temps show very temporally limited and shallow diurnal freeze only, and the contemporary active cryo-geomorphic environment has a negligible effect on the landscape/landforms today. However, the larger relict landforms

show us that this was indeed once a periglacial environment. So the caption could read something like 'Contemporary seasonal ice phenomena and relict periglacial landforms of the Bale Mnts'

Figure 5: It is problematic to show the location of only one needle ice site and only one frozen water fall locality. Firstly, there were likely other sites with needle ice at the time of observation...as also for frozen waterfalls or seepage out of rock at some localities. Secondly, the needle ice shown on the map is not a permanent feature at that locality, neither is the ice on the cliff face – which is hence problematic to show on a map. In Contrast, the other geomorphic phenomena mapped are permanent on the landscape (at least for the generation that will read this article) and thus suitable for mapping. Are the waterfalls (as shown in the photo) frozen every year? For how many months each year?

Figure 5b shows 3 exposure age locations but only one age given. '620' requires an indication of scale of age used. Why does the word saturated appear twice on the map? Is this not also a bit problematic...unless it is permanently saturated at that locality? Figure 5c three numeric values given......what are these ...age scale used?

P17, line 2: the authors say that the deposits are associated with so called 'frost weathering'. How do you know for certain that it was due to 'frost weathering' ...and not maybe a combination of different weathering mechanisms of which freezing/thawing of water might be one? This would then also imply potential thermal stress (thermoclastis) as an additional weathering type. I think greater scientific caution and rigor is required with statements such as these.

Figure 7: When I examine your temperature records over the period 2017 to 2019 in this Figure, I am concerned that your 2cm and 10cm ground temperature data may not actually represent the temperatures at a fixed depth through time because I can see that their amplitudes (in both the positive and negative directions) increases progressively through time. This is of course typical to a situation where your thermistor has

shifted upwards through the soil profile.

For temperature measurements and discussion re temperatures – why do you interchange between Kelvin and °C? Please keep to °C.

Way forward I think that the greatest strength this paper has to offer is in showcasing the very unique large sorted stripes and possibly large sorted patterned ground. Showcasing these features and finding a way to show their environmental significance (in a scientifically robust manner), surely merits publication, albeit as a much shorter article than the one submitted currently. I suggest a much trimmed down version of this paper: 1) briefly describing contemporary soil frost dynamics and small-scale contemporary soil frost phenomena – where some of the temperature data could be included, and 2) showcasing the large relict features with mapping data and field based measurement data (I currently do not see the value of the 36Cl and ground penetrating radar data). From these, one could then build an interesting but focused and concise discussion (along the lines of some of the discussion on p25, lines 17-31 – which I quite like). I caution against trying to make too much inference from relict landforms for which we still know relatively little in terms of their mechanisms of formation and thus underlying ground and air climatic requirements. It would thus not be possible to say too much about palaeo-climates for this region, let alone the tropics as a whole as the title of the paper implies. It might be worth saying something about the geo-heritage & geo-tourism potential here given the rarity/uniqueness of the landforms.

References: Boelhouwers, J., Holness, S., Meiklejohn, I., & Sumner, P. (2002). Observations on a blockstream in the vicinity of Sani Pass, Lesotho Highlands, southern Africa. Permafrost and Periglacial Processes, 13(4), 251-257. Harris, S. A., & Pedersen, D. E. (1998). Thermal regimes beneath coarse blocky materials. Permafrost and Periglacial Processes, 9(2), 107-120.

---

## Referee Comment (RC2) · Anonymous Referee #2 · 20 Jul 2020

General comments:

This paper presents a detailed account of current and past periglacial landforms and processes of the Bale Mountains in Ethiopia, with specific focus on relict sorted stone stripes. The latter is a very prominent feature and very unique for the tropics and mid- and high-latitudes in general. The characteristics of these stone stripes are described by detailed geomorphological mapping, UAV photogrammetry, ground-penetrating radar and 36Cl surface exposure dating. Palaeoclimatic importance are studied by collecting current ground and air temperature data and modelling a minimum air temperature depression needed to form these landforms.

[Figure]

I was pleased to receive the invitation for reviewing this work, which I read with great interest. It is clear to me that the authors have gathered a highly relevant dataset, which would be a great scientific contribution about this topic. However, I agree with previous referee report of Stefan Grab that the paper is very long, in some places lacking focus and it is not always clear what the added value of certain datasets are. For example the UAV photogrammetry data – is this just a nice addition or does it actively contributes to your findings? Are grain size distributions based on this imagery as you state in your methodology? This is not clear. I understand that the authors want to describe the features in as much detail as possible, but this does not come forward in the result section of the paper, where it seems that only the geomorphic mapping, the 36Cl surface exposure dating and the temperature measurements and modelling are presented. Results from UAV data and GPR seem to be lacking/could be stated more clearly. The way the paper is written now, the temperature measurements and analysis form the core of this work and all other methods are tributary. I strongly agree with Stefan Grab's suggestions on the temperature data used in this paper. The potential presence of air circulation in the blocky material, causing substantial cooling, should be discussed. In addition, comparison with current day examples could be more elaborate and is now only briefly touched in the discussion (on p25, L6 you state there are well documented examples from the high arctic). It is also not clear to me why the example of the Falkland Islands is highlighted. Is this the only other site that shows similar inactive landforms, like the ones you observe in the Bale Mountains?

Nevertheless, I also agree with Stefan Grab that this work is highly relevant and important to publish. I therefore suggest that a moderate to major revision of the manuscript is required.

Specific comments:

- Be careful with absolute statements that are not based on clear references/data. For example: P1 L1: . . .the most prominent features. . . -> one of the most/one of the more prominent features. . . (People studying rock glaciers might disagree with your

initial statement...). P2 L17: Africa's largest alpine environment -> one of Africa's largest alpine environment (also see comment of Stefan Grab). I see that in L22 on p3 underneath study area you have a more detailed statement of this, referring to your manuscript in revision. If you stand by this statement, consider moving this information more forward in the manuscript.

- You are very brief when describing the collected UAV data (L14-20 p6). Normally, at least an error reporting should be done to indicate the reliability of your data. Because UAV data is prone to deformation, especially when using a small amount of ground control points that might not be evenly distributed. If I understood correctly, you did not incorporate ground control points to process the images, but only to georeference the final products (orthophoto, DSM). This is confusing, since normally ground control points are used to correct the geometry in the 3D modelling procedure. Therefore, consider using different terminology. I understand that this is not the focus of your paper and that you refer to earlier work. However, I still think error reporting should be included here (and might not be similar as the errors you achieved on Kanderfirn) if you want to include this data in your paper.

- The text reads sometimes confusing when you talk about temperature measurements. Please check thoroughly throughout the document that you clearly mention when you talk about ground temperature and when you talk about air temperatures. For example: In the caption of figure 1: GT and TM are both ground temperature loggers?

- P13 L13: This sentence is lacking a reference. Since you base an important part of your modelling on this value, and the resulting temperature depression, you could give more attention to where you get this value. Is this -1°C ground temperature purely theoretical (from literature)? Or is this based on other observations in other areas? In your discussion you give an example of Goldthwait 1976, where large sorted landforms are common with air temperatures of -4 to -6°. How does this stand in relationship with -1°C ground temperature? Could you compare these air temperatures to the air temperatures depression you found for the Bale Mountains? This needs some clarification

throughout the document in both methodology, results and discussion.

- Section 4.1: make sure the distinction between active and relict periglacial processes is clear. Also add this to the title, for example . . . past and present periglacial processes (needle ice is not really a landform).

- Figure 5: I agree with Stefan Grab considering the comments about Figure 5. Reporting single frozen waterfalls and needle ice observation is rather anecdotal. Could you, besides direct observation, also indicate areas where these phenomena are likely, depending on elevation, slope, aspect. . .? Do you have more observations, from for example locals? You could make different mapping categories between permanent landforms and areas were current periglacial processes could be observed. Differ between landforms and processes.

- P16 L9: Is there a clear difference in elevation (belts) between relict periglacial features and current periglacial landforms/processes?

- P17, L1: Are the scree slopes really relict? Or could present frost weathering also still contribute to these landforms that are mainly formed in the past?

- P23 section 4.3: this section could use some rewriting. L9-10 contains your topic sentence, what this part is really about, and I would move this up to the beginning of your paragraph for clarity. At the end of this section you again state that -1°C MAGT seems critical for the formation of deep seasonal frost. On what do you base this statement? (see also previous comment).

- P23: Your discussion section could benefit greatly from adding subtitles. Now the structure is not clear and different things are discussed alternatingly, not always grouped coherently. The first paragraph reads more like a conclusion/summary. I can differ the following discussion topics from your current paragraphs:

Similar periglacial landforms in other areas/comparison of the Bale Mountains to other area (paragraphs 2-4) Specific environmental settings of the Bale Mountains (paragraph 5, but also 9 and 10) The formation of pattern ground (paragraph 6), discussion seasonal variation (paragraph 7) and sporadic permafrost (paragraph 8) Outreach and future research (paragraph 11)

Technical comments:

- The English of the paper can still be improved, especially long and complex sentence structures (e.g. multiple commas) should be avoided. Often readability already improves greatly if the sentence structure is reversed, or split into multiple phrases. For example: P2 L7-10 P3 L16-21: turn these two sentences around: The exact timing.. is unknown... due to lack of geological maps... The central Sanetti Plateau... is characteristic for the Bale Mountains. P3 L26-28 P17 L5-8

- Watch out with neglecting articles (a/an, the): P14 L25: the Sanetti Plateau, the highest peaks P20 L20: ...and the northern valleys...

- Take care of the use of hyphens: P1 L2 and P2 L28: mid- and high-latitudes

- Consider putting table 2 and figure 3 in Appendix.

- The use of allow: allow cannot be followed just by a verb, so things like "allow to establish" (L4 p12) are not correct. Allow needs either a noun or a subject and verb, like "allow the establishment of"

- Several times you refer to information that is stated later in the manuscript (for example L12, p13, L8, p15). This makes the structure of your paper not always clear to the reader. Consider moving important information more forward.

- p 17, L7: this sentence is already part of the next paragraph. Move for better structure.

- P22, L3: revise sentence, wrong use of minimal

- P23, L3: concurrently = simultaneously (?) – long sentence

- P23, L5: what = which

- P29 L27: Suggestion to add 'modern' and 'co-exist' :... where relict and modern, frost-related periglacial landforms co-exist..

- Spelling and grammar flaws are not all flagged, so careful proofreading is still required, keeping the above mentioned comments in mind.

- Figure captions should be clear independently from the text. Therefore, please clarify:

Figure 1: the control points, are they used for georeferencing the UAV data or for satellite imagery? GT and TM are both ground temperatures? The different figure panels require a, b, ... so the data basis can be referenced more clearly. Consider leaving out the map of Africa indicating the position of Ethiopia and instead mention in the text that Ethiopia is positioned in the horn of Africa. The map of Africa is lacking a scale, as well as your inset of the map of Ethiopia to show the position of the Bale Mts. )

Figure 5: Consider using a different color code for active and relict periglacial forms (and general geomorphology such as landslides). The distinction between stone polygons and stone nets are not clear, use a different symbol. Make it more clear that panel D is derived from the UAV data.

Figure 8: very long figure caption. Put the information of the columns into the figure (the months for Bega, Belg, Kiremt). No need for mentioning the colours for panel b, this is clear from the column headers. Specify if this displays air temperatures or ground temperature in the figure.

Figure 9: I assume this data is from the AWS stations, mention this clearly. Specify if this displays air temperatures or ground temperature in the figure.

Figure 10: I assume this data is from your loggers, mention this clearly. Specify if this displays air temperatures or ground temperature in the figure.

---

## Author Comment (AC1) · 29 Oct 2020

**General response**

First of all, we would like to thank the associate editor for obtaining two valuable reviews and the two referees, Stefan Grab and one anonymous, for their critical, thorough and constructive comments on our manuscript. We are pleased that both referees show interest in the presented data and findings and generally support the publication of the manuscript if their concerns are properly addressed. Both referees criticised the length of the manuscript and a lack of focus in some places. Hence, we decided to shorten and restructure the entire manuscript, to remove some of the ground temperature data, and change the initial title. We propose the following new title for the revised manuscript: *"The enigma of relict large sorted stone stripes in the tropical Ethiopian Highlands"*. The new title emphasizes the peculiarity of the studied stone stripes with respect to their size and occurrence on a tropical mountain. In the revised manuscript, we will focus on the characteristics of the stone stripes and outline two different scenarios regarding their genesis and age. We will keep some of the ground temperature data as well as the modelling experiment in the manuscript to discuss the recent climatic setting and elaborate on the potential palaeoclimatic and environmental conditions during the formation of the stone stripes. We are convinced that the manuscript will improve due to the reviewers' remarks, questions and suggestions. On the following pages, we respond to the reviewers' comments point by point. The reviewers' comments are highlighted in gray and the responses in white. We hope that the responses qualify us to submit a revised version of the manuscript.

**Response to Referee Comment 1 (Stefan Grab)**

Referee report on: Implications of present ground temperatures and relict stone stripes in the Ethiopian Highlands for the palaeoclimate of the tropics Authors: Groos et al.

General comments I am pleased to see that the Bale Mnts are receiving this geomorphic attention it deserves. I have worked on a variety of periglacial landforms in various African environments over many years now - but to me the large scale sorted stone stripes presented in this paper from the Sanetti Plateau, are the most special and unique periglacial landforms I have yet seen in Africa. They are truly special and must rank amongst the largest and best examples globally, I would think. So work on these is certainly called for and important to publish. I think the big challenge with these amazing landforms is ascertaining when they developed, how long they were actively forming for and when they might have become inactive (relict) periglacial phenomena. The second great challenge is ascertaining how they formed, because any misunderstanding as to their formation has serious implications to any climatic controls we attach to their genesis. I think these challenges are very real for this paper and when I read the work I see that this is where the paper has its struggles. I have several major concerns with this paper, which I will outline next, but at the same time wish to also assist with providing suggestions that might help rework this paper into something that might be publishable.

We thank Stefan Grab for his helpful comments and for his contribution to improve the manuscript. Below, we will respond to his major points of criticism.

Major concerns to address 1. The paper is far too long and tries to tackle too many things with too much detail, such that the connections between the various bits of collected data/information, become somewhat muddled and lost in the discussion/conclusion. The extent of detail to such things as instrumentation and story behind the logger battery issues etc may be valuable to place in a technical report or PhD thesis, but is not suitable for a journal publication. The text requires substantial trimming down and tightening up throughout.

We agree that some of the detailed ground temperature data and results go beyond the scope of the manuscript and may distract from the main discussion about the active periglacial processes and relict patterned grounds in the Bale Mountains. To improve the readability and structure of the manuscript, we decided to shorten and reorganise it. We will trim the entire text and remove some of the ground temperature data as well as figures 8-10 (others will be shifted to the appendix). Furthermore, we propose a new title for the revised version of the manuscript (see general remark). We will also include subheadings in the discussion as suggested by the anonymous referee.

2. Although the written style is relatively uncomplicated and for the most part satisfactory, there is a tendency towards colloquial language style, which is not suitable. The written style thus requires considerable improvement for publication.

We appreciate the general feedback regarding the written style and do not doubt that the language can still be improved, but it would have been helpful if some examples for the stated "tendency towards colloquial language" would have been provided. We will carefully proofread the manuscript again and revise the sentence structure and choice of words where necessary. We will also ask for professional language editing if that is recommended/requested for final publication.

3. a) The Scientific methods are notable and impressive for such a region given the logistical hurdles. However, great or large quantities of data may not always be the most useful or necessary

data for the study objectives. - While it is great having 36Cl results for the landforms, these raise more questions than provide answers. These do not necessarily inform us when the landforms first developed, or how long they were actively forming for, or when they became 'periglacially inactive'. So, despite all the efforts in obtaining exposure ages, the authors are still left with merely assuming that the landforms are of Late Glacial age. Such an assumption might be reached without the exposure ages and have in fact also been made for other openwork block deposits (e.g. block streams) in the high Drakensberg (southern Africa) – see for e.g. Boelhouwers et al. (2002). Without any real sense of timeframe, it is impossible to use the landforms for any palaeo-climate reconstructions.

We were aware that determining the stabilisation phase of relict periglacial landforms with cosmogenic nuclides like $^{36}$Cl is challenging and may be impossible if the sampled rocks have been exposed to cosmic radiation for several ten or hundred thousand years prior to (and/or during) their formation. However, several studies have demonstrated that the stabilisation phase of relict periglacial features like rock glaciers and block fields can be successfully dated with cosmogenic nuclides in some cases (e.g. Barrows et al., 2004; Ivy-Ochs et al., 2009; Steinemann et al., 2020). Since we were already sampling erratic boulders on the Sanetti Plateau for $^{36}$Cl surface exposure dating and the reconstruction of the former ice cap, we decided to test whether this method can also help to constrain the period when the stone stripes became inactive. As already discussed in the manuscript, this was apparently not the case. The sampled rocks reveal very high $^{36}$Cl concentrations, corresponding to exposure ages between 65 and 700 ka (recalculated with the latest version of the CRONUS Earth Web Calculator and a time-dependent "LSDn" scaling). The high concentrations could be interpreted in terms of a stone stripe formation over several cold stages during the Pleistocene as proposed for the stone runs in the Falkland Islands (Wilson et al., 2008). But they might also be the result of a long-term pre-exposure of the sampled rocks. Due to the well-preserved morphology and hardly weathered surface of the stone stripes as well as the required temperature depression, we argued that the stone stripes most likely formed during the coldest period(s) of the Late Pleistocene (i.e. MIS 2). The referee is right that such an assumption might be reached without the obtained exposure ages, but this is of course impossible to know before applying this method to the unique landform. Now that we have the $^{36}$Cl data, it would be odd not to present and discuss them (also from a methodological perspective). Furthermore, we think that the measured concentrations provide additional information that are valuable for the interpretation of the paleoenvironment. The high concentrations for example suggest (not necessarily prove) that the stone stripes were probably never covered by thick ice for thousands of years and might have formed at the margin of the paleo ice cap (all this is discussed in detail in the manuscript…). We will emphasize the added value of including the $^{36}$Cl dataset more clearly in the revised version of the manuscript.

3. b) Great effort was also undertaken with the Ground-penetrating radar measurements; something not previously done for periglacial landforms in Africa. However, the results to me do not show much that is of significance - and so does not add enough value to provide for anything noteworthy to add to the discussion or meet the aim/objectives of this paper. I would like to be proven wrong here – so if the authors can indeed use these data in a way that enhances/strengthens the discussion, then that would be good. - The authors provide considerable temperature data (ground and air). In fact I think too much is attempted with these temperature records and in the process of trying too

much with it (also too many graphs), the scientific value and merit is lost. I will elaborate on temperature data separately as this constitutes a major concern.

Information on the internal structure of the coarse stone stripes and fine regolith in between are essential to ascertain the genesis of this landforms and to address some important questions: How thick is the regolith layer above the volcanic bedrock? Is there any evidence of cryoturbation in the regolith layer? Does the regolith contain any larger blocks? Answering these questions would help to assess the former frost depth (or thickness of the active layer) and to evaluate whether frost heave and sorting were the main mechanisms for the formation of the stone stripes. Therefore, we intended to excavate a transect across the regolith (and if possible also across the coarse stone stripes). However, we didn't receive the necessary permission to do such an excavation in the national park. As an alternative, we conducted a first GPR survey. The GPR measurements show a relatively homogeneous regolith layer without any major reflection. The radargram is visually not very "exciting", but it provides a minimum depth for the regolith layer of more than 1.5 m. In addition, the GPR survey suggests that the regolith layer between the stone stripes contains no larger rocks (i.e. exceeding several decimetres). We interpret this observation, the coexistence of alternating regolith and coarse stine stripes, as an indication of frost sorting, which is one important mechanism and precondition for the formation of sorted patterned grounds. We will stress in the revised manuscript that these data are noteworthy for the discussion of the genesis of the stone stripes.

4. Temperature data: While the temperature data recorded at various localities might be used for various scientific purposes, I think the way in which the data have been used in this paper requires very careful reconsideration. - The work is built upon the presumption that the sorted stripes are a product of past seasonal or sporadic permafrost that would have required ground temps of -1∘ C . . .or a thermal reduction of around 12∘ C from those recorded more recently. And the authors argue on their modelling basis that air temps would thus have been lowered by around 7.6∘ C. This is all highly presumptuous and very controversial. In the first instance, ground temperatures were measured in the finer textured soil stripes and not within the openwork block deposits. When these features first formed, they may have formed in a scree of such open-work block deposition because of unique localized air flow (cooling) with depth through such openwork material, thus possibly creating 'pockets' of long lasting frozen ground phenomena (be it extended seasonal freeze or permafrost). So, the soils in which the authors have done their measurements may not have been as extensively frozen as for instance in the adjacent blocky material. Please see some published work which has shown enhanced cooling through blocky periglacial phenomena (e.g. Harris & Pedersen (1998) show much colder ground thermal conditions below blocky materials than finer textured regolith cover). - A further point is that the authors have not considered the likely thermal impacts of snow in a palaeo-environmental context. It is thus impossible to begin modelling likely air temperature reductions unless we know 1) the actual palaeo-ground temperature at exactly the same site as the contemporary measurements were taken (which is assumed to have been below 0∘ C – but very much built on assumption as the stripes may have formed when there was deep freeze beneath the blocky material but only limited/shallow/seasonal freeze in the finer textured soil stripes), and 2) the depth, duration etc of snow which would have had an insulating effect – or maybe helped preserve cooling during particular times of the year etc. The distribution and thickness of snow across the landscape would almost certainly have had impacts on the

spatial/temporal characteristics of ground freeze and thaw during past colder periods. In summary re the temperature data – it is 'stretching the data too far' to try and start modelling past air temperatures as the scientific context is far too simplistic in the way it has been presented here. In reality, the contexts are much more complicated than the authors make it out to be. At best, I think the authors can use contemporary ground temperature data to reflect on contemporary shallow soil frost phenomena.

We do not share the referees concern regarding the ground temperature modelling experiment and are convinced that it is an appropriate method to assess which paleoclimatic conditions would have theoretically been required for the formation of the stone stripes on the Sanetti Plateau. First of all, we would like to point out that "experimental and numerical modelling of Earth surface processes" is one of the journal's main subject areas. Hence, we think that such a modelling experiment generally fits the scope of journal. Based on all evidence we have gathered so far, the structure and characteristics of the stone stripes can be best explained by the mechanism of frost heave and sorting. Although it does not seem very likely, we can of course not exclude that a different, yet unknown, geomorphic mechanism might have caused the formation of the stone stripes. But if we assume that sub-freezing air temperatures and ground frost (no matter how deep) were an essential paleoclimatic precondition for their genesis (e.g. Goldthwait, 1976; Washburn, 1980), we can use our recent measurements to provide an order of magnitude for a past ground and air temperature depression in the Bale Mountains. The modern mean annual air temperature at the location of the stone stripes (~3900 m) is ~6 °C and the mean ground temperature (in the regolith) ~11 °C. The seasonal air and ground temperature minima are ~4 °C and 9 °C, respectively. This would translate into an air and ground temperature depression of at least ~4-6 °C and 9-11 °C, respectively. We agree at this point with the referee that the stone stripes might have formed at the beginning in a scree of open-work block deposits that promoted colder ground thermal conditions than the fine textured regolith layer (we pick up on this topic in a response further below...). However, this does not affect our assumption on the potential air temperature depression because sub-freezing air temperatures are probably also a precondition for the formation of frost in such open-work block deposits. During further sorting of the material, also the regolith between the stone stripes must have experienced frost action. Otherwise the absence of any larger blocks in the fine regolith between the coarse stone stripes is hard to explain. Hence, we think that the postulated rough ground temperature depression of at least 9-11 °C is justified. The referee also raised concern that the thermal impacts of snow on ground temperature were neglected. He argues that snow might have "helped preserve cooling during particular times of the year". The distribution and thickness of snow certainly modulates the energy exchange between the atmosphere and ground. Snow has an isolating effect and may prevent the ground from cooling as well as heating, depending on the timing of snowfall. However, the absence of any glacial landforms or sediments on the southern plateau margin as well as along the southern escarpment suggests that the area of the stone stripes was very dry during the Late Pleistocene. Larger quantities of snow should have promoted the glaciation of the southern and western Sanetti Plateau, but this was apparently not the case (see Ossendorf et al., 2019; Groos et al., under review). Apart from that, a seasonal snow cover by itself would have required (or led to) a ground temperature at the freezing point or below. Without a decrease in ground temperature of more than 9-11 °C, neither seasonal nor annual frost could have formed on the Sanetti Plateau. We did not claim anywhere in the manuscript that we intended to

simulate the energy exchange between the ground and atmosphere on the plateau in a physical manner, where the consideration of the snow pack etc. would have been indispensable. What we wanted to test with the simple statistical model is under which paleoclimatic conditions (i.e. decrease in temperature and insolation) the mean ground temperature would approximate frost conditions. An essential assumption underlying such an approach is that the established statistical relationship between the dependent and independent variables did not change. This is not necessarily the case as the thermal conductivity and heat capacity of the ground might for example change due to variations in soil moisture. We will discuss the limitations of our model experiment in more detail in the revised version of the manuscript. Furthermore, we will interpret the results in a more conservative way.

References:

- Barrows, T., Stone, J. O., and Fifield, L. K.: Exposure Ages for Pleistocene Periglacial Deposits in Australia, Quat. Sci. Rev., 23, 697–708, https://doi.org/10.1016/j.quascirev.2003.10.011, 2004.
- Goldthwait, R. P.: Frost Sorted Patterned Ground: A Review, Quat. Res., 6, 27–35, 1976.
- Ivy-Ochs, S., Kerschner, H., Maisch, M., Christl, M., Kubik, P. W., and Schlüchter, C.: Latest Pleistocene and Holocene Glacier Variations in the European Alps, Quat. Sci. Rev., 28, 2137–2149, https://doi.org/10.1016/j.quascirev.2009.03.009, 2009.
- Steinemann, O., Reitner, J. M., Ivy-Ochs, S., Christl, M., and Synal, H.-A.: Tracking rockglacier evolution in the Eastern Alps from the Lateglacial to the early Holocene, Quat. Sci. Rev., 241, 1–19, https://doi.org/10.1016/j.quascirev.2020.106424, 2020.
- Washburn, A. L.: Permafrost features as evidence for climatic change, Earth-Sci. Rev., 15, 327–402, https://doi.org/10.1016/0012-8252(80)90114-2, 1980.

More detailed technical matters to address: P3, line 23: How do you define 'alpine environment' . . .on what basis? Is it based on a Eurocentric view of 'alpine', or is it based on what has commonly been defined as the 'Afro-alpine' zone? I am not advocating any given view but the authors should define what they understand makes the Bale Mnts the largest African 'alpine' environment. . .as opposed to for instance the Atlas Mnts or high Drakensberg-Maloti mnt system in Lesotho, Southern Africa (in both these cases one might argue for extensive 'alpine and/or Afro-alpine' regions which are larger than that of the Bale).

We refer to Hedberg (1951), who defined the afro-alpine belt in tropical Africa as the area above ~3500 m. Others set the lower elevation of the tropical afro-alpine belt to ~3200 m (e.g. de Deus Vidal Jr & Clark 2019). However, we are aware that the afro-alpine belt tends to be lower in the subtropical mountains like the High Atlas or Drakensberg. The afro-alpine area on these mountains might therefore be similar or larger than that of the Bale Mountains. To be more specific, we will write "With an area of more than 1000 km$^2$, the Bale Mountains comprise Africa's most extensive alpine environment above 3500 m". Furthermore, we will directly refer in the text to the references below.

References:

- Hedberg O.: Vegetation belts of East African Mountains. Svensk bot Tidskr, 45, 140–202, 1951.

minimal- de Deus Vidal Jr, J. and Clark V. R.: Afro-Alpine Plant Diversity in the Tropical Mountains of Africa.Encyclopedia of the World's Biomes, 373–394, https://doi.org/10.1016/B978-0-12-409548-9.11885-8, 2019.

P5, lines 13-14- the values of glacial extent mentioned here is according to who? Needs a reference.

We refer to our separate manuscript on the glacial chronology of the Bale Mountains that is currently under review (Groos et al., in review). The reference will be included after the statement.

P5, line 20: the authors say here that the large periglacial features are associated with freeze-thaw processes. Unless the authors can verify that they have measured freezing and thawing dynamics here, and that these mechanisms produced these landforms, then this is a scientific assumption. So rather write as '. . ..features are likely associated with. . ..'

As outlined above, we think it is a reasonable scientific assumption that the stone stripes are the product of frost heave and sorting. However, we cannot provide any direct evidence for this assumption as the stone stripes constitute an inactive landform. Hence, we will write "The formation of these features is likely associated with freeze-thaw processes [...]".

P 5, line 30 would read better to say are 'endemic to'

The concerning section will be removed to shorten the manuscript.

P13, line 11: Stone stripes apparently required a thick active layer. Why do you say it had to be thick? What do you understand to be 'thick' rather than 'thin'? What dimensions are we dealing with here? Can it be that the relict sorted stripe sorting depth might say something re to active layer thickness. . .or depth to which [periglacial] geomorphic mechanisms operated?

With "thick" we meant an active layer that was at least several decimetres thick (as mentioned before and afterwards in the manuscript). We used the attribute "thick" to differentiate between decimetre to metre deep freezing and thawing, which we assume initiated the past formation of the large sorted stone stripes (see response to general comments), and superficial ("thin") frost that creates the present small scale patterned grounds on the Sanetti Plateau. We will specify this in the revised version of the manuscript.

Whether the depth (~2 m) of the relict sorted stone stripes says something about the active layer thickness (or depth to which the freezing front operated in case of the absence of permafrost) is an interesting and important question. The largest clasts of the stone stripes are up to 2 m long, 0.5 m wide, and weigh probably more than 1000 kg. We assume that an active layer or freezing depth of at least several decimetres was necessary to heave and sort these large clasts. The freezing front usually descends faster in coarse blocks than in fine material (Kessler and Werner, 2003). The faster descend of the freezing front in coarse blocks is mainly explained by three different processes: 1) The better thermal coupling between the ground and air due to enhanced conduction (i.e. heat transfer) through the blocks in the presence of snow (Juliussen and Humlun, 2008) . 2) Pronounced ground cooling through free convection of air in coarse blocks in the absence of snow (Wicky and Hauck, 2020). 3) Coarse blocks retain less water than wetter fine-grained soils that must be cooled before freezing (Kessler and Werner, 2003). Due to these mechanisms, it is likely that the freezing front at the interface between the coarse stone stripes and the fine regolith was inclined (causing lateral frost heave), while it was rather horizontal in the regolith (causing vertical frost heave), coresponding to the concept presented by Kessler and Werner (2003). Since the freezing front was probably deeper below the coarse and trough-shaped stone stripes than in the fine and rampart-like

regolith (mimicking the morphology of the landform), the relative frost depth (distance between ground surface and freezing front) was probably smaller than the sorting depth (~2 m) of the stone stripes, but larger than a few decimetres (as inferred from the size of the clasts). This would provide a potential frost depth (or active layer thickness) in the order of several decimetres up to 2 m during the formation period of the stone stripes. Cryoturbation was probably limited to a depth of maximum 2-3 m as this seems to be the maximum regolith thickness on the plateau.

References:

- Juliussen, H. and Humlun, O.: Thermal regime of openwork block fields on the mountains Elgåhogna and Sølen, central-eastern Norway, Permafrost Periglac. Process., 19, 1–18, http://doi.wiley.com/10.1002/ppp.607, 2008.
- Kessler, M. A. and Werner, B. T.: Self-Organization of Sorted Patterned Ground, Science, 299, 380–383, https://doi.org/10.1126/science.1077309, 2003.
- Wicky, J. and Hauck, C.: Air Convection in the Active Layer of Rock Glaciers, Front. Earth Sci., 8, 1–17, https://www.frontiersin.org/article/10.3389/feart.2020.00335/full, 2020.

P14 – at the bottom of this page the authors list so called 'frost-induced phenomena' such as frozen waterfalls, needle ice, patterned ground and solifluction lobes. This is a bit confusing as it mixes geomorphic periglacial landform types (i.e. patterned ground and solifluction lobes) with ice types (massive ice as frozen waterfalls or needle ice developed in soil). Ground ice types might be seen as mechanistic agents, while the landforms might be seen as products of the former.

In the revised version of the manuscript, we will distinguish more carefully between ice types and landforms.

Figure 4: These are impressive photos and all valuable to add here. In photo g, I can see the patterned ground (blocky borders) – in fact they look impressive to me, but the dotted white line that the authors have placed to supposedly outline the borders (shape) do not correspond with the pattern border localities in the photo.

Thanks, we realised that the simplified/idealised shape of the drawn borders does not align with the actual borders of the patterns shown in Fig. 4g. We will remove the anticipated (dotted) borders in the revised figure.

The caption to Figure 4 is a bit misleading I think. It informs the reader that these photos show us the 'Periglacial environment of the Bale Mnts'. In the first instance, it shows contemporary phenomena of a frozen waterfall and needle ice (i.e. the contemporary environment). These features do not qualify this to be labelled a contemporary periglacial environment as the ground temps show very temporally limited and shallow diurnal freeze only, and the contemporary active cryo-geomorphic environment has a negligible effect on the landscape/landforms today. However, the larger relict landforms show us that this was indeed once a periglacial environment. So the caption could read something like 'Contemporary seasonal ice phenomena and relict periglacial landforms of the Bale Mnts'

We will follow this suggestions and modify the caption accordingly.

Figure 5: It is problematic to show the location of only one needle ice site and only one frozen water fall locality. Firstly, there were likely other sites with needle ice at the time of observation. . .as also for frozen waterfalls or seepage out of rock at some localities. Secondly, the

needle ice shown on the map is not a permanent feature at that locality, neither is the ice on the cliff face – which is hence problematic to show on a map. In Contrast, the other geomorphic phenomena mapped are permanent on the landscape (at least for the generation that will read this article) and thus suitable for mapping. Are the waterfalls (as shown in the photo) frozen every year? For how many months each year?

The location marked with an asterisk in the map (Fig. 5a) is predestined for cold air ponding and one of the best sites in the Bale Mountains to observe/study needle ice. Needle ice occurs also along other stream banks, but during our field surveys at the end of the dry season (January to February) most of the smaller streams were already dried up. In the upper Wasama Valley, needle ice can be found at clear nights throughout the dry season each year. That's why this location was included in the map.

As far as we have observed, the frozen waterfalls only evolve at the shaded north-exposed cliffs in the Wasama Valley (marked in the map). Our local guides were not aware of any other location where this seasonal phenomenon can be observed. The waterfalls in the Wasama Valley freeze every year at the end of the rainy season (i.e. October/November) and persist until the onset of the following wet season (i.e. February/March).

We will revise this figure (and caption) to clearly distinguish between the location of present and relict landforms as well as sites where seasonal phenomena like the frozen waterfalls and needle ice can be (best) observed.

Figure 5b shows 3 exposure age locations but only one age given. '620' requires an indication of scale of age used. Why does the word saturated appear twice on the map? Is this not also a bit problematic. . .unless it is permanently saturated at that locality? Figure 5c three numeric values given. . .....what are these . . .age scale used?

This information is indeed a bit misleading. The word "saturated" appeared twice in the map (5b) since $^{36}$Cl reached saturation (concentration where production and decay are balanced) in two samples of the investigated stone stripe on the western part of the plateau when using a time-invariant scaling (i.e. Stone, 2000). The resulting age (>1000 ka) of these two samples (BS05 and BS06) was at the limit of the method and was therefore not explicitly stated. As indicated in the legend, the three numeric values in Fig. 5c represent the exposure ages of three investigated blocks (BS01-03) from a stone stripe on the southern part of the plateau. We will add the sample names in the revised version of the map and provide further information in the caption for clarification. Furthermore, we will recalculate all six exposure ages (samples BS0-06) with the latest version (2.1.) of the established CRONUS Earth Web Calculator (http://cronus.cosmogenicnuclides.rocks/2.1/html/cl/) using the physics-based and time-dependent LSDn scaling framework (Lifton et al., 2014; Marrero et al. 2016). This information will be included in the respective methods section and in the figure caption.

References:
- Lifton, N., Sato, T. and Dunai, T. J.: Scaling in situ cosmogenic nuclide production rates using analytical approximations to atmospheric cosmic-ray fluxes. Earth Planet. Sci. Lett., 386, 149–160, http://dx.doi.org/10.1016/j.epsl.2013.10.052, 2014.
- Marrero, S. M., Phillips, F. M., Borchers, B., Lifton, N., Aumer, R. and Balco, G.: Cosmogenic nuclide systematics and the CRONUScalc program. Quat. Geochronol., 31, 160–

187, http://dx.doi.org/10.1016/j.quageo.2015.09.005, 2016.
- Stone, J. O.: Air pressure and cosmogenic isotope production. Journal of Geophysical Research: Solid Earth, 105, 23753–23759, http://doi.wiley.com/10.1029/2000JB900181, 2000.

P17, line 2: the authors say that the deposits are associated with so called 'frost weathering'. How do you know for certain that it was due to 'frost weathering' . . .and not maybe a combination of different weathering mechanisms of which freezing/thawing of water might be one? This would then also imply potential thermal stress (thermoclastis) as an additional weathering type. I think greater scientific caution and rigor is required with statements such as these.

We wrote that these deposits are "[…] associated inter alia with frost weathering […]". The large size of the clasts (several decimetres to more than one metre) indicates that frost action was probably the most relevant mechanism for producing the scree. However, this statement does not exclude that also other weathering processes were involved. We will rephrase the sentence to make this clear.

Figure 7: When I examine your temperature records over the period 2017 to 2019 in this Figure, I am concerned that your 2cm and 10cm ground temperature data may not actually represent the temperatures at a fixed depth through time because I can see that their amplitudes (in both the positive and negative directions) increases progressively through time. This is of course typical to a situation where your thermistor has shifted upwards through the soil profile.

It is unclear whether the referee refers to figure panel "a" or "b" in Fig. 7 and how he comes to the conclusion that the daily ground temperature amplitude increases progressively over the period 2017 to 2020.

Panel "a" in Fig. 7 shows hourly ground temperatures and, thus, allows to infer daily amplitudes. However, the analysis of the evolution of the daily ground temperature amplitude in 2 cm depth shows no significant trend over the period 2017-2020 (see figure below) and also no change after the readout dates (vertical dashed lines). If the last week before the final readout end of January 2020 was ignored in the analysis (this period was characterised by exceptionally high amplitudes), the minimal (insignificant) positive trend would even reverse into a minimal (insignificant) negative trend. This argues against the assumption that the thermistors might have shifted upwards through the profile over time. There is no evidence that the measured ground temperatures do not represent the conditions at fixed depths ($2 \pm 1$, $10 \pm 2$, $50 \pm 5$ cm).

Panel "b" in Fig. 7 shows seasonal ground temperature variations in 2, 10 and 50 cm depth and indicates that the thermal difference between the dry and wet seasons was more pronounced in the years 2018 and 2019 than in 2017. However, these variations are attributable to climatic fluctuations and cannot indicate a vertical shift of the thermistor as the magnitude of seasonal ground temperature variations is similar in all depth. To make this clear, we will use a simpler smoothing approach than the initial "local regression with a smoothing span of 0.32" to illustrate seasonal ground temperature variations in Fig. 7b.

[Figure]

For temperature measurements and discussion re temperatures – why do you interchange between Kelvin and ◦C? Please keep to ◦C.

We used °C for actual temperatures and Kelvin for temperature differences (as it is common in engineering sciences) to avoid misinterpretations, but since this caused confusion, we will consistently use °C for actual temperatures and temperature differences in the revised version of the manuscript.

Way forward I think that the greatest strength this paper has to offer is in showcasing the very unique large sorted stripes and possibly large sorted patterned ground. Showcasing these features and finding a way to show their environmental significance (in a scientifically robust manner), surely merits publication, albeit as a much shorter article than the one submitted currently. I suggest a much trimmed down version of this paper: 1) briefly describing contemporary soil frost dynamics and small-scale contemporary soil frost phenomena – where some of the temperature data could be included, and 2) showcasing the large relict features with mapping data and field based measurement data (I currently do not see the value of the 36Cl and ground penetrating radar data). From these, one could then build an interesting but focused and concise discussion (along the lines of some of the discussion on p25, lines 17-31 – which I quite like). I caution against trying to make too much inference from relict landforms for which we still know relatively little in terms of their mechanisms of formation and thus underlying ground and air climatic requirements. It would thus not be possible to say too much about palaeo-climates for this region, let alone the tropics as a whole as the title of the paper implies. It might be worth saying something about the geo-heritage & geo-tourism potential here given the rarity/uniqueness of the landforms.

As outlined in the responses above, we will shorten and restructure the manuscript. The new structure would be as follows:

Results

4.1 Contemporary ground frost dynamics and small-scale ground frost phenomena (including some ground temperature data from the Sanetti Plateau)

4.2 Characteristics of the relict periglacial features (including the $^{36}$Cl and GPR data as discussed in the responses above).

4.3 Results of the ground temperature modelling experiment

Discussion

5.1 Comparison of the stone stripes in the Bale Mountains with similar landforms in other regions

5.2 Genesis and age of the stone stripes (discussing two different scenarios: 1. formation due to seasonal temperature variations, 2. formation over several cold and warm phases during the Pleistocene)

5.3 Stone stripes as proxy/indication for regional paleoclimatic and environmental changes

We will shortly discuss the geo-heritage and geo-tourism potential in 5.1.

References: Boelhouwers, J., Holness, S., Meiklejohn, I., & Sumner, P. (2002). Observations on a blockstream in the vicinity of Sani Pass, Lesotho Highlands, southern Africa. Permafrost and Periglacial Processes, 13(4), 251-257. Harris, S. A., & Pedersen, D. E. (1998). Thermal regimes beneath coarse blocky materials. Permafrost and Periglacial Processes, 9(2), 107-120.

Thanks for these recommendations. Will consider both references for the discussion in the revised manuscript.

**Response to Referee Comment 2 (anonymous)**

General comments:

This paper presents a detailed account of current and past periglacial landforms and processes of the Bale Mountains in Ethiopia, with specific focus on relict sorted stone stripes. The latter is a very prominent feature and very unique for the tropics and mid- and high-latitudes in general. The characteristics of these stone stripes are described by detailed geomorphological mapping, UAV photogrammetry, ground-penetrating radar and 36Cl surface exposure dating. Palaeoclimatic importance are studied by collecting current ground and air temperature data and modelling a minimum air temperature depression needed to form these landforms.

I was pleased to receive the invitation for reviewing this work, which I read with great interest. It is clear to me that the authors have gathered a highly relevant dataset, which would be a great scientific contribution about this topic. However, I agree with previous referee report of Stefan Grab that the paper is very long, in some places lacking focus and it is not always clear what the added value of certain datasets are.

We thank the anonymous referee for his helpful comments and for his contribution to improve the manuscript. In the following, we will respond to the general comments.

As outlined in the general response at the beginning and in the response to Stefan Grab, we decided to shorten and reorganise the manuscript in order to improve the readability and structure. We will trim the entire text and remove some of the ground temperature data as well as figures 8-10 (others will be shifted to the appendix). Furthermore, we propose a new title for the revised version of the manuscript (see general response). We will also include subheadings in the discussion as you suggested in one of your comments below.

For example the UAV photogrammetry data – is this just a nice addition or does it actively contributes to your findings? Are grain size distributions based on this imagery as you state in your methodology? This is not clear. I understand that the authors want to describe the features in as much detail as possible, but this does not come forward in the result section of the paper, where it seems that only the geomorphic mapping, the 36Cl surface exposure dating and the temperature measurements and modelling are presented. Results from UAV data and GPR seem to be lacking/could be stated more clearly.

The UAV photogrammetry data are important for the manuscript due to the following reasons: i) The 3D aerial image composed of multiple UAV images (Fig. 6) provides the most realistic view of the stones stripes and helps the reader to get an impression of the landform. ii) The digital surface model served as basis for the analysis of the stone stripes morphology. iii) The orthophoto served for measuring the size of the clasts in the coarse stone stripes. To emphasise the added value of each individual dataset, we will state more clearly in the results section of the revised manuscript how each dataset actively contributes to our main findings.

The way the paper is written now, the temperature measurements and analysis form the core of this work and all other methods are tributary. I strongly agree with Stefan Grab's suggestions on the temperature data used in this paper. The potential presence of air circulation in the blocky material, causing substantial cooling, should be discussed.

Regarding the usage of the ground air temperature date, we would like to redirect the referee to our general response (4.) to the other referee Stefan Grab.

In addition, comparison with current day examples could be more elaborate and is now only briefly touched in the discussion (on p25, L6 you state there are well documented examples from the high arctic). It is also not clear to me why the example of the Falkland Islands is highlighted. Is this the only other site that shows similar inactive landforms, like the ones you observe in the Bale Mountains?

Nevertheless, I also agree with Stefan Grab that this work is highly relevant and important to publish. I therefore suggest that a moderate to major revision of the manuscript is required.

We discussed the stone stripes in the Bale Mountains in comparison with the "stone runs" in the Falkland Islands since this is the only other known location worldwide where inactive stone stripes of similar size have been reported and described. In the revised manuscript, we will dedicate one subchapter to the comparison of the stone stripes in the Bale Mountains with past and present-day examples elsewhere.

Specific comments:
- Be careful with absolute statements that are not based on clear references/data. For example: P1 L1: . . .the most prominent features. . . -> one of the most/one of the more prominent features. . . (People studying rock glaciers might disagree with your initial statement. . .). P2 L17: Africa's largest alpine environment -> one of Africa's largest alpine environment (also see comment of Stefan Grab). I see that in L22 on p3 underneath study area you have a more detailed statement of this, referring to your manuscript in revision. If you stand by this statement, consider moving this information ore forward in the manuscript.

We will be more careful with absolute statements and will rephrase the first sentence in the abstract. The Bale Mountains represent indeed the largest afro-alpine environment above 3500 m (for a more detailed explanation please refer to the response to the other referee), but we didn't explicitly mention the elevation in the quoted sentence. We will modify this.

- You are very brief when describing the collected UAV data (L14-20 p6). Normally, at least an error reporting should be done to indicate the reliability of your data. Because UAV data is prone to deformation, especially when using a small amount of ground control points that might not be evenly distributed. If I understood correctly, you did not incorporate ground control points to process the images, but only to georeference the final products (orthophoto, DSM). This is confusing, since normally ground control points are used to correct the geometry in the 3D modelling procedure. Therefore, consider using different terminology. I understand that this is not the focus of your paper and that you refer to earlier work. However, I still think error reporting should be included here (and might not be similar as the errors you achieved on Kanderfirn) if you want to include this data in your paper.

We neglected a more detailed description since the general processing procedure of the UAV images was already presented in a previous publication (Groos et al., 2019), although with a focus on a completely different region (the Swiss Alps...). The absolute georeferencing accuracy of the orthophoto and DSM in not of major relevance in our case as we do not compare our data with any other datasets. However, we agree that an error estimate must always be reported. The horizontal (XY) accuracy of the provided dataset is ~0.5 m (relative to the orthorectified WorldView-1 image) and the vertical (Z) accuracy is ~1.5. m (relative to the SRTM-1 DEM). We will include this

information in the revised manuscript. We had no differential GPS available with us and, thus, could not directly mark and measure ground control points in the field. In principle, the number of five "relative" control points that we used to process the images and georeference the final products are sufficient for such a small area (60 x 80 m) (e.g. Gindraux et al., 2017). Warping (often referred to as "doming" and "fishbowling" effect) can be a problem when processing UAV images, but this is rather the case for larger study areas and not for such a small area as we surveyed. We will revise the respective method section about the UAV data and explain the procedure more precisely.

Reference:
- Gindraux, S., Boesch, R., and Farinotti, D.: Accuracy Assessment of Digital Surface Models from Unmanned Aerial Vehicles' Imagery on Glaciers. Remote Sensing, 9, 1–15, http://www.mdpi.com/2072-4292/9/2/186, 2017.
- Groos, A. R., Bertschinger, T. J., Kummer, C. M., Erlwein, S., Munz, L., and Philipp, A.: The Potential of Low-Cost UAVs and Open-Source Photogrammetry Software for High-Resolution Monitoring of Alpine Glaciers: A Case Study from the Kanderfirn (Swiss Alps), Geosciences, 9, 1–21, https://doi.org/10.3390/geosciences9080356, 2019.

- The text reads sometimes confusing when you talk about temperature measurements. Please check thoroughly throughout the document that you clearly mention when you talk about ground temperature and when you talk about air temperatures. For example: In the caption of figure 1: GT and TM are both ground temperature loggers?

With a few exceptions, we have explicitly distinguished between air and ground temperature throughout the manuscript, but in the caption of Fig. 1 we forgot to mention it. We will revise the caption and will check the manuscript carefully again to differentiate between air and ground temperature where we haven't so far.

- P13 L13: This sentence is lacking a reference. Since you base an important part of your modelling on this value, and the resulting temperature depression, you could give more attention to where you get this value. Is this -1 ◦C ground temperature purely theoretical (from literature)? Or is this based on other observations in other areas? In your discussion you give an example of Goldthwait 1976, where large sorted landforms are common with air temperatures of -4 to -6 ◦. How does this stand in relationship with -1 ◦C ground temperature? Could you compare these air temperatures to the air temperatures depression you found for the Bale Mountains? This needs some clarification throughout the document in both methodology, results and discussion.

As far as we are aware, active stone stripes of similar size like those in the Bale Mountains and Falkland Islands do not exist elsewhere on the globe (or have at least not been reported/described until today). Hence, the mean annual ground temperature that was prevailing during the formation of the stone stripes is difficult to assess. If we assume that frost heave and sorting was one precondition for the formation of the stone stripes, the mean annual ground temperatur at that time was probably about 0 °C or below in the Bale Mountains. Seasonal minimum ground temperatures below 0 °C (and an MAGT above 0°C) might have been theoretically sufficient for the stone stripe formation, but due to the size of the clasts and minor seasonal temperature variations in the tropics, we think a MAGT of 0 °C is an adequate first assumption. Ground temperatures oscillating about 0 °C are also typical for regions where stone circles form (see for example Hallet 2013). The MAGT might have been lower in the Bale Mountains, but it is impossible to make any specific and

justified assumption. Therefore, we will replace the initially used, but not well-founded, theoretical value of -1 °C by a MAGT of 0 °C. We will justify our assumption and clarify our procedure (as outlined above) throughout the revised manuscript.

Reference:
- Hallet, B.: Stone Circles: Form and Soil Kinematics, Proc. R. Soc. A, 371, 20120 357, https://doi.org/10.1098/rsta.2012.0357, 2013.

- Section 4.1: make sure the distinction between active and relict periglacial processes is clear. Also add this to the title, for example . . . past and present periglacial processes (needle ice is not really a landform).

We will split the results section (4.) into a subchapter on active (4.1) and a subchapter on relict (4.2) periglacial processes and landforms to avoid confusion.

- Figure 5: I agree with Stefan Grab considering the comments about Figure 5. Reporting single frozen waterfalls and needle ice observation is rather anecdotal. Could you, besides direct observation, also indicate areas where these phenomena are likely, depending on elevation, slope, aspect. . .? Do you have more observations, from for example locals? You could make different mapping categories between permanent landforms and areas were current periglacial processes could be observed. Differ between landforms and processes.

We (and also our local guides) didn't observe frozen waterfalls at any other location. We mentioned this phenomenon in the text (and marked it in the map) because it seems to be the only apparent frost-related process in the Bale Mountains, which is controlled by seasonal rather than diurnal temperature fluctuations. We will add additional locations where needle ice typically occurs (based on observations of our colleagues). Furthermore, we will modify the legend and caption of this figure to clearly distinguish between permanent landforms (active and relict) and areas where periglacial processes can be observed.

- P16 L9: Is there a clear difference in elevation (belts) between relict periglacial features and current periglacial landforms/processes?

No, there is not clear difference in elevation (belts) between the relict periglacial features and current periglacial landforms/processes. What we mainly observed is a difference in the magnitude of the landforms/processes (e.g. active small-scale patterned grounds vs. sorted stone stripes and polygons on the Sanetti Plateau).

- P17, L1: Are the scree slopes really relict? Or could present frost weathering also still contribute to these landforms that are mainly formed in the past?

We cannot exclude that present weathering still contributes to (some of) these landforms, but the return of Erica shrubs/trees between the stones as well as the lack of parent material (i.e. cliffs) at some locations indicates that they mainly formed in the past. We will modify the sentence to make this clear.

- P23 section 4.3: this section could use some rewriting. L9-10 contains your topic sentence, what this part is really about, and I would move this up to the beginning of your paragraph for clarity. At the end of this section you again state that -1 ∘ C MAGT seems critical for the formation of deep seasonal frost. On what do you base this statement? (see also previous comment).

We will restructure and rewrite this paragraph. A MAGT of 0 °C (or previously -1 °C) seems critical for the formation of several decimtre deep frost and the genesis of the stone stripes because of the following two reasons: 1. If the MAGT was above 0 °C (e.g. 1 or 2 °C), seasonal frost could probably not form in several decimetre depth due to limited seasonal ground temperature variations (in the order of only ± 1 °C). 2. If the MAGT was below 0 °C (e.g. -1 or -2 °C), seasonal (or permanent) frost could probably not thaw to a depth of several decimetre due to limited seasonal ground temperature variations (in the order of only ± 1 °C).

If the stone stripes formed not because of seasonal ground temperature variations (oscillating around 0°C), but due to temperature fluctuations over several cold (mean multi-annual ground temperatures <0 °C) and warm periods (mean multi-annual ground temperatures >0 °C) during the Pleistocene, the magnitude of temperature change between these phases seems decisive. Will will discuss both scenarios in the revised manuscript.

- P23: Your discussion section could benefit greatly from adding subtitles. Now the structure is not clear and different things are discussed alternatingly, not always grouped coherently. The first paragraph reads more like a conclusion/summary. I can differ the following discussion topics from your current paragraphs: Similar periglacial landforms in other areas/comparison of the Bale Mountains to other area (paragraphs 2-4) Specific environmental settings of the Bale Mountains (paragraph 5, but also 9 and 10) The formation of pattern ground (paragraph 6), discussion seasonal variation (paragraph 7) and sporadic permafrost (paragraph 8) Outreach and future research (paragraph 11)

We will restructure the discussion and consider your suggestion to include subtitles. The first part (5.1) will focus on the comparison between the large sorted stone stripes in the Bale Mountains and similar landforms in other regions around the globe. The second part (5.2) will focus on the genesis and age of the stone stripes and the third part (5.3.) will discuss the potential of the stone stripes as proxy/indication for regional paleoclimatic and environmental changes.

Technical comments:
- The English of the paper can still be improved, especially long and complex sentence structures (e.g. multiple commas) should be avoided. Often readability already improves greatly if the sentence structure is reversed, or split into multiple phrases. For example: P2 L7-10 P3 L16-21: turn these two sentences around: The exact timing.. is unknown. . . due to lack of geological maps. . . The central Sanetti Plateau. . . is characteristic for the Bale Mountains. P3 L26-28 P17 L5-8

Thanks for the feedback regarding the written style. We will carefully proofread the manuscript again and revise the sentence structure and choice of words where necessary. We will also ask for professional language editing if that is recommended/requested for final publication.

- Watch out with neglecting articles (a/an, the): P14 L25: the Sanetti Plateau, the highest peaks P20 L20: . . .and the northern valleys. . .

We will check the manuscript and add articles where they are missing.

- Take care of the use of hyphens: P1 L2 and P2 L28: mid- and high-latitudes

We will add hyphens where they are missing.

- Consider putting table 2 and figure 3 in Appendix.

We will remove Fig. 3. Table 2 will be trimmed considerably due to the removal of about two thirds

| of the ground temperature dataset. |
|---|
| - The use of allow: allow cannot be followed just by a verb, so things like "allow to establish" (L4 p12) are not correct. Allow needs either a noun or a subject and verb, like "allow the establishment of" |
| Thanks, we will correct this throughout the manuscript. |
| - Several times you refer to information that is stated later in the manuscript (for example L12, p13, L8, p15). This makes the structure of your paper not always clear to the reader. Consider moving important information more forward. |
| We will restructure the manuscript accordingly. |
| - p 17, L7: this sentence is already part of the next paragraph. Move for better structure. |
| Will move this sentence. |
| - P22, L3: revise sentence, wrong use of minimal |
| We will revise this sentence. |
| - P23, L3: concurrently = simultaneously (?) – long sentence |
| We will rephrase this sentence. |
| - P23, L5: what = which |
| Will be corrected. |
| - P29 L27: Suggestion to add 'modern' and 'co-exist' :. . . where relict and modern, frost-related periglacial landforms co-exist. |
| Will be changed. |
| - Spelling and grammar flaws are not all flagged, so careful proofreading is still required, keeping the above mentioned comments in mind. |
| As aforementioned, we will carefully proofread the manuscript to eliminate the remaining flaws. |
| - Figure captions should be clear independently from the text. Therefore, please clarify:

Figure 1: the control points, are they used for georeferencing the UAV data or for satellite imagery? GT and TM are both ground temperatures? The different figure panels require a, b, . . . so the data basis can be referenced more clearly. Consider leaving out the map of Africa indicating the position of Ethiopia and instead mention in the text that Ethiopia is positioned in the horn of Africa. The map of Africa is lacking a scale, as well as your inset of the map of Ethiopia to show the position of the Bale Mts. |
| We will modify the figure caption to explain more clearly the meaning of the different labels/datasets (control points, GT and TM ground tempeature data logger, etc.). Control points were used for processing and georeferencing the UAV data. GT and TM are the high-quality and low-cost ground temperature data logger, respectively. We will label the different figure panels. We will leave the map of Africa out and include a scale bar. |
| Figure 5: Consider using a different color code for active and relict periglacial forms (and general geomorphology such as landslides). The distinction between stone polygons and stone nets are not clear, use a different symbol. Make it more clear that panel D is derived from the UAV data. |

We will consider all suggestions and revise the figure and caption accordingly.

Figure 8: very long figure caption. Put the information of the columns into the figure (the months for Bega, Belg, Kiremt). No need for mentioning the colours for panel b, this is clear from the column headers. Specify if this displays air temperatures or ground temperature in the figure.

This is a helpful suggestion, but we decided to remove this figure and the corresponding data to shorten the manuscript.

Figure 9: I assume this data is from the AWS stations, mention this clearly. Specify if this displays air temperatures or ground temperature in the figure.

Yes, the data originate from the weather stations, but we forgot to mention this in the caption. It is explicitly stated in the caption that air temperature lapse rates are displayed. However, this figure will be removed to narrow the focus of the manuscript.

Figure 10: I assume this data is from your loggers, mention this clearly. Specify if this displays air temperatures or ground temperature in the figure.

It is explicitly mentioned in the figure caption that ground temperatures are displayed and that the data originate from the installed loggers. However, this figure and the corresponding data will be removed to trim the manuscript.

---

## Editor Comment (EC1) · Heather Viles (Editor) · 30 Oct 2020

I have read the reviewers' comments and the author's response carefully and recommend that the authors revise the manuscript in the way they have outlined in their response. Once resubmitted it will be sent to the reviewers for their reassessment.

---

## Author Response (AR2)

**General response to the managing editor, associate editor, and referee**

We thank Tom Coulthard and Heather Viles for their positive feedback and for accepting our paper for publication in Earth Surface Dynamics. We also acknowledge Stefan Grab's effort to read the revised manuscript once again. His critical and constructive remarks during the review greatly enhanced the structure and quality of the paper.

**Specific response to minor technical issues raised by Stefan Grab (Referee 1)**

| |
|---|
| Dear Authors
This manuscript is substantially improved and I commend you for it. In my view this is now an excellent paper certainly worthy of publication. Just a few small technical issues to sort:
1. The term 'patterned ground' is widely accepted within periglacial science as a collective term for the patterns concerned and hence the plural that you use in this paper seems strange and incorrect (i.e patterned grounds). Almost all papers on patterned ground refer to it as such and the plural form used in your submitted paper would not conform. Please change to 'patterned ground' throughout. |
| Thanks for the clarification. We now use the term "sorted patterned ground" in singular form throughout the paper. |
| 2. Figure 4 part b - I understand that this part of the Figure displays 'seasonal' temperature patterns as you say in the caption. In fact the seasonal pattern is very beautifully shown! However, the resolution is much finer than only seasonal...it is at least monthly. Hence the data resolution does not comply with the resolution implied (i.e. seasonal) in the caption. The same issue applies with Figure 7. So this requires the captions to be changed accordingly...maybe something like '.....indicating monthly to seasonal temperature resolution......'. Please also check the text re compliance in this regard. |
| We agree with the referee that the temporal resolution is actually higher than implied in the caption. We have therefore changed the captions of Fig. 4 and Fig. 7 accordingly:
***Figure 4.*** *Hourly ground temperatures in 2, 10, and 50 cm depth on the southern Sanetti Plateau (3877 m) from January 2017 to January 2020. (b) Smoothed hourly ground temperatures using a simple moving average with a window size of 91 days, highlighting seasonal ground temperature variations. Note that the increase of the seasonal ground temperature amplitude over the measurement period is also confirmed for other sites on the plateau and is not caused by a shift of the thermistors.*
***Figure 7.*** *Simulated daily mean ground temperatures in (a) 2 cm, (b) 10 cm, and (c) 50 cm depth on the southern Sanetti Plateau (3877 m) corresponding to a decrease in air temperature of 7.1 ± 1.3 °C and a decrease in global radiation of 30 W/m² relative to the present-day conditions. (d) Smoothed daily ground temperatures using a simple moving average with a window size of 91 days, highlighting seasonal ground temperature variations.* |
| 3. Page 15, line 5: '...stripes split up into multiple....' is a colloquial read. Perhaps consider something like '.....stripes diverge into multiple branches....' |
| We modified the sentence following the referee's suggestion. |